# Interpretable Graph Embeddings: Feature-Level Decomposition for Trustworthy Graph Neural Networks

## Abstract

Graph Neural Networks (GNNs) have achieved state-of-the-art performance on tasks such as user-item interaction prediction in recommender systems, molecular property classification, and credit risk scoring and fraud detection in financial risk modeling. However, their opaque embedding mechanisms raise critical concerns about transparency and trustworthiness. Existing explainability approaches largely focus on identifying the nodes, edges, or subgraphs that influence the model's prediction but fail to disentangle how individual node features shape learned embeddings. In this work, we propose a novel decomposition framework that systematically attributes each embedding to original node and/or edge features. We qualitatively demonstrate the framework on Graph Convolutional Networks (GCN) and Heterogeneous GraphSAGE (HinSAGE) using Cora and MovieLens, and quantitatively benchmark against widely adopted baselines across multiple datasets. Results indicate that our approach improves interpretability by revealing how node features contribute to individual graph embeddings and clarifying the role of neighborhood aggregation in shaping predictions.This work connects structural explainability and feature-level attribution, providing a principled foundation for trustworthy and actionable GNN explanations.[1]

## 1 Introduction

Graph Neural Networks (GNNs) have established themselves as powerful models for relational and structured data, achieving state-of-the-art performance in fields ranging from molecular property prediction to recommender systems to financial risk analysis. Such models leverage both the features of the entities represented by nodes and the connections between them, as well as the structure of the graph created by these relationships. However, their black-box nature has raised concerns about transparency and accountability, spurring a growing literature on explainability of GNNs. Recent surveys (Yuan et al., 2023; Kakkad et al., 2023) provide comprehensive taxonomies of this work, categorizing methods into post-hoc attribution techniques, model-specific explainers, and inherently interpretable architectures. Despite this progress, transparently explaining the node feature information captured in GNN embeddings remains an open question.

Much of the early work focused on structural explanations, identifying which nodes, edges, or subgraphs are most influential for a given prediction. Methods such as GNNExplainer (Ying et al., 2019) and PGExplainer (Luo et al., 2020) learn masks to highlight critical subgraphs, while approaches like SubgraphX (Yuan et al., 2021) employ Monte Carlo search to locate and extract task-relevant structures. Perturbation-based methods extend this idea by quantifying the importance of nodes or edges through controlled modifications of the graph. These approaches have proven valuable in identifying important structures, but do not provide insight into the node features captured by the GNNs.

In parallel, researchers have adapted gradient- and propagation-based methods from computer vision, such as Integrated Gradients (IG), Layer-wise Relevance Propagation, and GraphLRP, to trace

---

[1]The views expressed in this paper are solely those of the authors and do not necessarily reflect the views of their affiliated institutions.

attribution signals back through GNN layers(Pope et al., 2019; Baldassarre & Azizpour, 2019; Schnake et al., 2021). These methods provide finer-grained insights into node features, yet they often conflate structural and feature importance due to message passing. Moreover, their dependence on local gradient signals makes them highly sensitive to noise, unstable under small perturbations, and prone to correlation bias. More principled frameworks, including counterfactual and causal explanations, have emerged to assess how hypothetical perturbations affect predictions. For instance, CF-GNNExplainer (Lucic et al., 2022) perturbs adjacency matrices to find the minimal perturbation to the input graph such that the prediction changes. However, these approaches also remain primarily concerned with structural or instance-level contributions rather than the feature-level decomposition of embeddings.

More recent work has expanded the landscape. For example, DEGREE (Feng et al., 2023) decomposes GNN mechanisms to attribute predictions to subgraph components, and D4Explainer (Chen et al., 2023) introduces in-distribution explanations through diffusion-based counterfactuals. DyExplainer (Wang et al., 2023) extends interpretability to dynamic GNNs by capturing temporal dependencies, while GraphOracle (Du et al., 2025) provides self-explainable class-level subgraphs without requiring post-hoc search. Other advances include FIGNN (Raut et al., 2025), which emphasizes feature-specific interpretability. At the same time, evaluation frameworks like GraphXAI (Agarwal et al., 2023) have benchmarked existing explainers, while robustness studies have highlighted their fragility to adversarial perturbations. In terms of interpretability of node embeddings, Dalmia & Gupta (2018) first analyzed how embedding dimensions correlate with basic graph properties, revealing implicit structural signals. Piaggesi et al. (2024) introduced DINE, which restructures embeddings for dimensional interpretability, ensuring each dimension reflects meaningful substructures. Extending this idea, Piaggesi et al. (2025) developed a disentangled, self-explainable representation learning approach that enforces semantic separation across dimensions.

Despite these advancements, there remains a lack of methods that systematically decompose a target node's final embedding into contributions from individual features of the nodes, edges, and their neighbors. In applications where speed or infrastructure concerns are critical, learned graph embeddings may be used to capture the information learned by a GNN and fed into downstream predictive systems. There, they act as engineered features to boost performance over models with only observed features. These embeddings capture the underlying structural relationships and feature interactions within the graph, summarizing multi-hop dependencies and relational patterns into compact representations, incorporating the influence of neighbors on outcomes. While this practice produces clear performance gains, it also raises accountability challenges: if embeddings drive decisions in sensitive contexts such as credit risk assessments, fraud detection, or medical diagnosis, then stakeholders should be able to trace which node and edge features shaped these embeddings and to what degree. For instance, in a fraud detection scenario, graph embeddings may increase a model's ability to detect fraud, but it is critical to understand which properties of the consumer, transaction, and merchant were captured by the GNN in order to understand evolving fraud patterns and develop mitigation strategies.

Motivated by this gap, this paper introduces a novel method for feature-wise decomposition of embeddings, enabling fine-grained attribution that complements structural explanations. By explicitly accounting for correlations among features, our method provides a faithful mechanism to trace how information is transformed through GNN layers into the target embedding representation. This reveals what node information is captured by the graph embeddings, thereby aligning predictive power with the demands of accountability and interpretability in high-stakes domains. Table 1 summarizes the capabilities of representative GNN explanation methods. While prior approaches can attribute predictions to node features or edges, they generally do not explain embeddings, and most rely on optimization, sampling, or architecture-specific constraints. Our decomposition framework is unique in directly tracing embeddings back to raw features, supporting aggregation across embeddings, and producing deterministic, efficient attributions through simple matrix multiplications.

The remainder of this work is structured as follows. In Section 2, we present our decomposition framework for inverting embedding generation in GNNs and demonstrate its application on two representative architectures: Graph Convolutional Network (GCN) and Heterogeneous GraphSAGE (HinSAGE). Section 3 illustrates the approach empirically using the Cora citation network (Sen et al., 2008) for node classification to demonstrate feature-wise decomposition for GCN embeddings, and the MovieLens dataset (Harper & Konstan, 2015) for link regression to highlight type-aware and

| Capability | GNNExplainer | IG | LIME/GraphLIME | PGExplainer | GraphSVX | GOAt | FIGNN | Ours |
|---|---|---|---|---|---|---|---|---|
| Explain node features | ✓ | ✓ | ✓ | × | ✓ | ✓ | ✓ | ✓ |
| Explain embeddings directly | × | × | × | × | × | × | × | ✓ |
| Aggregate across embedding dims. | × | × | × | × | × | × | × | ✓ |
| Post-hoc on trained models | ✓ | ✓ | ✓ | ✓ | ✓ | ✓ | × | ✓ |
| Deterministic (no sampling) | × | × | × | × | × | ✓ | ✓ | ✓ |
| Handles high-dim features | × | ✓ | × | ✓ (edges) | × | △ | △ | ✓ |
| Domain flexibility | ✓ | ✓ | ✓ | ✓ | ✓ | △ | △ | ✓ |

Table 1: Capability comparison of representative GNN explanation methods. Symbols: ✓ = supported; × = not supported; △ = partially supported.

edge-level explanations in HinSAGE. Section 5 discusses implications, limitations, and potential extensions of our method. Finally, Section 6 concludes the paper.

## 2 METHOD

We propose a framework for decomposing graph neural network (GNN) embeddings into feature-wise contributions. The central observation is that, once the nonlinearity from the architecture is fixed (e.g., ReLU with a given input), each GNN layer becomes a linear operator for that input. This allows us to propagate contributions of node and edge features through successive layers and exactly reconstruct each output embedding as a sum over these features. We illustrate the framework with two widely used architectures: the Graph Convolutional Network (GCN) (Kipf & Welling, 2017) for node classification and Heterogeneous GraphSAGE (HinSAGE) (Hamilton et al., 2017; Ying et al., 2018; Zhang et al., 2019) for heterogeneous link prediction. For clarity, we summarize the notation used in Table 3.

### 2.1 GENERAL FRAMEWORK

A message-passing GNN layer that uses the relational graph convolutional operator (Schlichtkrull et al., 2018) can be expressed as follows:

$$\mathbf{h}_v^{(\ell)} = \sigma\Big( W_{\text{self}}^{(\ell)} \mathbf{h}_v^{(\ell-1)} + \sum_{r \in \mathcal{R}} W_r^{(\ell)} \mathcal{A}_r(\{\mathbf{h}_u^{(\ell-1)} : u \in N_r(v)\}) + \mathbf{b}^{(\ell)} \Big),$$

where $\mathcal{A}_r$ is a *linear aggregator* (e.g., normalized sum, mean, or sampled mean) and $\mathcal{R}$ indexes edge types or relations.

To invert this process, we propagate *contribution matrices* in parallel to the forward pass. Initialization is $\mathbf{C}_{w \to v}^{(0)} = \mathbf{x}_w^\top$ if $w = v$, and 0 otherwise. Propagation is then

$$\mathbf{C}_{\cdot \to v}^{(\ell+1)} = D_v^{(\ell)}\Big( W_{\text{self}}^{(\ell)} \mathbf{C}_{\cdot \to v}^{(\ell)} + \sum_{r \in \mathcal{R}} W_r^{(\ell)} \mathcal{A}_r(\{\mathbf{C}_{\cdot \to u}^{(\ell)} : u \in N_r(v)\}) + \mathbf{b}^{(\ell)} \Big),$$

where $D_v^{(\ell)}$ is a diagonal matrix that encodes the activation pattern for the given input for node $v$ at layer $\ell$. After $L$ layers, $\mathbf{h}_v^{(L)}$ can be exactly decomposed as a sum over $\{\mathbf{C}_{w \to v}^{(L)}\}$, with each term corresponding to a source node $w$ and input feature dimension $p$. Contribution vectors can be summarized into scalar importance scores using norms, and can also be projected for visualization using a PCA-based method (see Appendix for details).

### 2.2 GRAPH CONVOLUTIONAL NETWORKS (GCN)

The Graph Convolutional Network (GCN) (Kipf & Welling, 2017) is an adjacency-based GNN widely used for semi-supervised node classification. For two-layer GCN, each layer applies normalized adjacency $\tilde{A} = D^{-\frac{1}{2}}(A + I)D^{-\frac{1}{2}}$ to mix neighbor features:

$$\mathbf{H}^{(1)} = \sigma\big(\tilde{A}XW^{(0)} + \mathbf{b}^{(0)}\big), \quad \mathbf{H}^{(2)} = \sigma\big(\tilde{A}\mathbf{H}^{(1)}W^{(1)} + \mathbf{b}^{(1)}\big).$$

For a target node $v$, the contribution from feature $p$ of node $w$ to the embedding $\mathbf{h}_v^{(2)}$ is

$$\left[\mathbf{C}_{w\to v}^{(2)}\right]_{p,:} = D_v^{(1)}\left(\sum_{u\in\mathcal{V}} \tilde{A}_{vu}\, D_u^{(0)}\big(\tilde{A}_{uw}\, \mathbf{x}_w[p]\, \mathbf{e}_p^\top W^{(0)} + \mathbf{b}^{(0)}\big)W^{(1)}\right). \tag{1}$$

This expansion shows that signals propagate along paths $w \to u \to v$. Self-contributions arise when $w = v$, first-hop contributions when $w \in N(v)$, and two-hop contributions when $w$ connects via some $u$. Grouping terms provides hop-level or self/neighbor breakdowns of the learned representation.

## 2.3 HETEROGENEOUS GRAPHSAGE (HINSAGE)

GraphSAGE (Hamilton et al., 2017) is a widely used inductive GNN framework that learns node embeddings by sampling and aggregating information from each node's neighborhood. Hin-SAGE (Ying et al., 2018; Zhang et al., 2019) generalizes GraphSAGE to heterogeneous graphs with multiple node and edge types. Instead of operating with a full adjacency matrix, it samples fixed-size neighborhoods per hop, stratified by type. Each neighbor type has its own projection matrix. For node $v$ of type $t_0$ at layer $\ell$:

$$\mathbf{h}_v^{(\ell)} = \sigma\Big(W_{t_0,\text{self}}^{(\ell)}\mathbf{h}_v^{(\ell-1)} + \sum_{t\in\mathcal{T}} W_{t\to t_0}^{(\ell)}\, \frac{1}{|N_t(v)|} \sum_{u\in N_t(v)} \mathbf{h}_u^{(\ell-1)} + \mathbf{b}_{t_0}^{(\ell)}\Big).$$

Our decomposition naturally extends:

$$\mathbf{C}_{\cdot\to v}^{(\ell+1)} = D_v^{(\ell)}\Big(W_{t_0,\text{self}}^{(\ell)}\mathbf{C}_{\cdot\to v}^{(\ell)} + \sum_{t\in\mathcal{T}} W_{t\to t_0}^{(\ell)}\, \tfrac{1}{|N_t(v)|} \sum_{u\in N_t(v)} \mathbf{C}_{\cdot\to u}^{(\ell)} + \mathbf{b}_{t_0}^{(\ell)}\Big). \tag{2}$$

Because contributions are partitioned by node type, we obtain explanations such as "merchant features" vs. "account features," reflecting the heterogeneous semantics. For link prediction, HinSAGE produces edge embeddings $\psi(\mathbf{h}_u^{(L)}, \mathbf{h}_v^{(L)})$, commonly via Hadamard product or concatenation, followed by a linear classifier. Since these operators are linear in $\mathbf{h}_u^{(L)}$ and $\mathbf{h}_v^{(L)}$, contributions extend seamlessly. For example, with Hadamard product:

$$\ell_{uv} = \mathbf{w}^\top(\mathbf{h}_u^{(L)} \odot \mathbf{h}_v^{(L)}) + c,$$

the contribution of feature $p$ of node $w$ is obtained by combining node-level contributions with $\text{diag}(\mathbf{w})\mathbf{h}_v^{(L)}$ or $\text{diag}(\mathbf{w})\mathbf{h}_u^{(L)}$, depending on whether $w$ lies in the neighborhood of $u$ or $v$. This yields edge-level decompositions that directly attribute predicted links to original features of source and destination neighborhoods. Because HinSAGE uses random neighborhood sampling, explanations are conditional on the computation graph. Averaging across samples produces expected contributions, while a single sample yields instance-specific explanations.

By expressing GNN layers as masked linear operators and propagating contributions in parallel to the forward pass, our framework provides exact, activation-conditioned decompositions of embeddings into original features. The GCN case highlights hop- and neighbor-wise propagation, while HinSAGE showcases type-aware, edge-level sampling in heterogeneous graphs. Together, these examples demonstrate our method's generalizability across major GNN architectures, enabling principled, feature-level interpretability of embeddings and predictions.

## 3 QUALITATIVE EXPERIMENTS

### 3.1 DATASETS AND EXPERIMENTAL SETUP

To illustrate the proposed decomposition framework, we consider two datasets using two GNN architectures: the Cora citation network (Sen et al., 2008) using GCN and the MovieLens dataset (Harper & Konstan, 2015) using HinSAGE. Further details are provided in the Appendix. All models are implemented using the StellarGraph library (Data61, 2018).

## 3.2 RESULTS

**Cora (GCN case study).** We compare an XGBoost model (Chen & Guestrin, 2016) trained on raw 1,433 BoW features against an XGBoost model trained on only the 16 GCN embeddings from the last hidden GCN layer. Using embeddings yields a performance lift, improving accuracy from 0.57 to 0.76 and weighted F1 score from 0.56 to 0.75. Since the embeddings are predictive features for downstream tasks, explaining the embeddings is necessary to understand the information captured by the GCN.

We decompose each embedding back to the original features by expanding the actual passing with the trained weights and ReLU gates. For a target node $v$, source node $w$, and feature index $p$, the contribution vector $\left[\mathbf{C}^{(2)}_{w \to v}\right]_{p,:}$ can be calculated using the equation 1, where $D^{(0)}$ and $D^{(1)}$ denote the diagonal ReLU gating at layer 1 and 2. This makes the two-hop paths explicit and preserves the exact trained computation (e.g., bias flow and ReLU gates). In all feature-attribution summaries below, we exclude the bias term so that values reflect word contributions only. To separate whether an embedding's influence originates from the node itself or from neighbors, we decompose the renormalized adjacency into diagonal and off-diagonal parts,

$$\tilde{A} = \tilde{A}_{\text{self}} + \tilde{A}_{\text{nbr}}, \qquad (\tilde{A}_{\text{self}})_{uw} = \begin{cases} \tilde{A}_{uu}, & u = w, \\ 0, & u \neq w, \end{cases}.$$

Let $\left[\mathbf{C}^{(2)}_{w \to v}\right]_{p,:}$ denote the layer-2 contribution vector from feature $p$ of source node $w$ to the embedding of target node $v$. We obtain *self-origin* and *neighbor-origin* contributions by replacing the factor $\tilde{A}_{uw}$ in that expansion with $(\tilde{A}_{\text{self}})_{uw}$ and $(\tilde{A}_{\text{nbr}})_{uw}$, respectively: $\left[\mathbf{C}^{(2)}_{\text{self}, w \to v}\right]_{p,:} :=$ $\left[\mathbf{C}^{(2)}_{w \to v}\right]_{p,:}\Big|_{\tilde{A}_{uw} \leftarrow (\tilde{A}_{\text{self}})_{uw}}$, and $\left[\mathbf{C}^{(2)}_{\text{nbr}, w \to v}\right]_{p,:} := \left[\mathbf{C}^{(2)}_{w \to v}\right]_{p,:}\Big|_{\tilde{A}_{uw} \leftarrow (\tilde{A}_{\text{nbr}})_{uw}}$. Since embeddings can exhibit correlation, we optionally apply a PCA-based projection to visualize attribution patterns in an orthogonal basis (see Section A.2). This step is intended for interpretability and visualization. Let $V \in \mathbb{R}^{d \times r}$ be the top $r$ PCA loadings (columns orthonormal) fit once on $\mathbf{H}^{(2)}$. We rotate each contribution vector over the embedding axis and sum the first $r$ principal components:

$$\widetilde{\mathbf{C}}^{(2)}_{\star, w \to v}[p, 1{:}r] = \mathbf{C}^{(2)}_{\star, w \to v}[p, :]\, V, \qquad \left[\mathbf{s}^{\text{PCA}}_{\star}(v)\right]_p = \sum_{w \in \mathcal{V}} \sum_{c=1}^{r} \widetilde{\mathbf{C}}^{(2)}_{\star, w \to v}[p, c], \quad \star \in \{\text{self}, \text{nbr}\},$$

where $d = 16$ and $r = 5$ in this example. Stacking rows over $v$ yields matrices

$$S_{\text{self}},\ S_{\text{nbr}} \in \mathbb{R}^{N \times F},$$

whose $v$-th rows are the correlation-adjusted word attributions $\mathbf{s}^{\text{PCA}}_{\text{self}}(v)^{\top}$ and $\mathbf{s}^{\text{PCA}}_{\text{nbr}}(v)^{\top}$.

To visualize contributions, we apply t-SNE (Maaten & Hinton, 2008) to reduce $S_{\text{self}}$ and $S_{\text{nbr}}$ (1,433 words) to two dimensions. Figure 1 shows contributions obtained by decomposing GCN embeddings, comparing self-node and neighbor-node representations. Our intention is not to claim strong separation based solely on t-SNE, but rather to provide an intuitive illustration of contribution patterns. Incorporating neighborhood information appears to influence embedding structure, which aligns with observed gains in recall performance (e.g., *Reinforcement Learning*: 0.44 to 0.72, *Theory*: 0.18 to 0.56, *Rule Learning*: 0.37to 0.64). Figure 3 in the appendix reports full performance results across all categories. Because this analysis is conducted at the feature-contribution level, it offers transparency into what each embedding learns individually by grouping features with similar contribution behaviors, helping interpret the role of neighborhood aggregation in shaping model predictions.

**MovieLens (HinSAGE case study).** We then interpret the embeddings generated by the HinSAGE model. Following the approach used for the GCN example, we treat these embeddings as additional features and compare two XGBoost models: one using only user and movie attributes, and another incorporating both raw attributes and embeddings. Incorporating embeddings improves predictive performance across key metrics: MSE decreases from 1.12 to 0.95, MAE from 0.86 to 0.78, and $R^2$ nearly doubles from 0.11 to 0.24. Feature importance from the XGBoost regressor (total gain) highlights the predictive value of embeddings, with only one raw user attribute (scaled age) appearing among the top 20 features. These results underscore the necessity of explaining

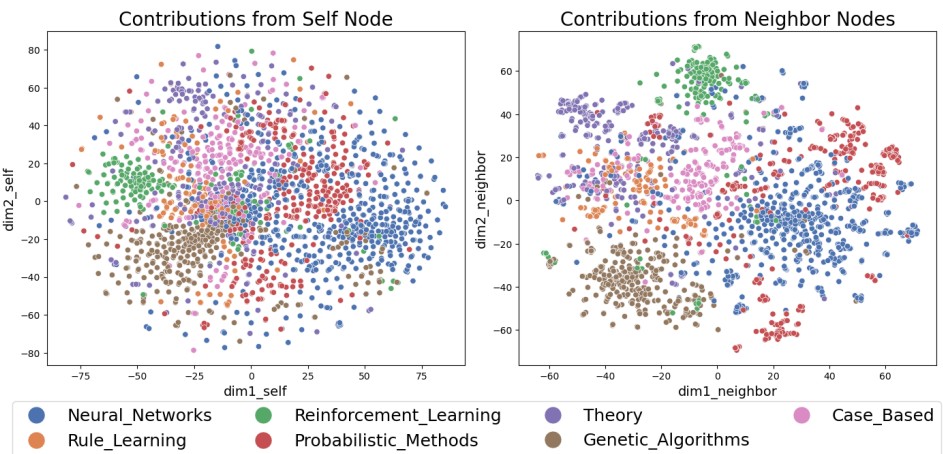

Figure 1: t-SNE visualization of feature contributions from decomposed GCN embeddings. The left panel shows contributions from self nodes, while the right panel shows contributions from neighbor nodes. Notably, the neighbor node contributions exhibit clearer separation across categories, indicating their stronger role in capturing class-discriminative information.

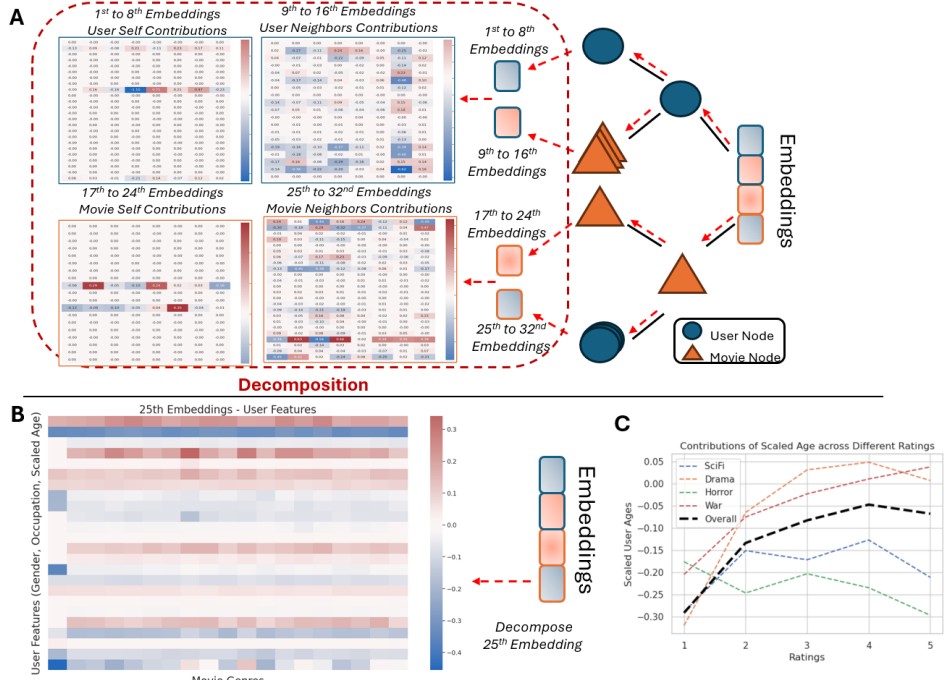

Figure 2: A. Decomposition of user–movie HinSAGE embeddings into self-contributions and neighbor-contributions for both users and movies, illustrating how individual components construct the learned representation. B. Analysis of the $25^{th}$ embedding dimension by aggregating contributions across user groups and comparing them across different movie genres, highlighting systematic differences in representation structure. C. Contributions of the user attribute *scaled age* across different rating levels, aggregated by movie genres.

embeddings, as they encode highly informative signals. Figure 4 in the appendix provides the full comparisons between two XGB models across different rating levels.

To this end, we decompose the learned embeddings using the equation 2. Figure 2A illustrates this process for a single user–movie pair.Then, we leverage learned weights to map each embedding back to its contributing features. The decomposition yields four contribution matrices corresponding to: (a) user self-node (dimensions 1–8), (b) user's neighboring movie nodes (9–16), (c) movie self-node (17–24), and (d) movie's neighboring user nodes (25–32). For example, the unnormalized value of the 17th embedding dimension is $-0.18$, with the movie attributes *drama* and *horror* contributing $-0.06$ and $-0.12$, respectively, while other attributes have negligible impact. Beyond single-pair analysis, we examine the most influential embedding ($25^{th}$, with an importance score of 0.18 using the "total_gain" metric) by aggregating contributions across multiple pairs. As shown in Figure 2A, this embedding primarily captures information from neighboring users of the target movie, reflecting collaborative signals. Figure 2B compares aggregated contributions across user groups and movie genres, revealing systematic differences in representation structure (e.g., $-0.21$ for *animation* vs. 0.09 for *documentary*). Finally, we analyze the role of a specific user attribute—*scaled age*—across rating levels and genres. Figure 2C shows that *war* and *drama* genres exhibit trends aligned with the overall population (increasing with rating), whereas *sci-fi* and *horror* display the opposite pattern, with younger users tending to assign higher ratings. Contribution values amplify these differences: *war* and *drama* show near-zero contributions, while *sci-fi* and *horror* exhibit strongly negative contributions, indicating their distinct influence on embedding formation.

Appendix A.4 provides additional results. Figure 5 shows average scaled-age contributions to the $25^{th}$ embedding; Figure 6 examines the `job=artist` subpopulation across rating levels. High-contribution genres trend upward with rating, whereas low-contribution genres are essentially flat. In addition, we analyze the $11^{th}$ embedding—constructed from users' movie-neighbor signals—with a focus on the *War* genre (Figure 7); compared to contribution results from $25^{th}$ embedding dimension, the contribution curves for the $11^{th}$ embedding cluster more tightly, indicating that differences among features are subtle and not strongly discriminative for this genre. These findings demonstrate that our proposed attribution-based decomposition provides a step toward improving interpretability of graph-based embeddings and offers insights into how heterogeneous relational signals influence predictive performance.

## 4 QUANTITATIVE EXPERIMENTS

### 4.1 DATASETS AND EXPERIMENTAL SETUP

We conducted quantitative experiments to compare our explanation method against a suite of baselines, including GOAt (Lu et al., 2024), GNN-LRP (Schnake et al., 2021), Integrated Gradients (IG) (Sundararajan et al., 2017b), GradCAM (Pope et al., 2019), LIME (Ribeiro et al., 2016), and a Random baseline. All methods operate on the same GCN backbone under identical training configurations to ensure fairness. Specifically, the backbone consists of 2 graph convolutional layers with ReLU activation, followed by a linear output layer. Each hidden layer has 64 dimensions, and training is performed for 500 epochs using the Adam optimizer with a learning rate of 0.05 and weight decay of $5 \times 10^{-4}$. A dropout rate of 0.5 is applied to mitigate overfitting.

**Evaluation Metrics.** We adopt faithfulness-based metrics widely used in interpretability research and adapted for graph settings (Yuan et al., 2023). Let $f(\cdot)$ denote the GNN classifier that outputs the predicted probability for the ground-truth class, $x$ the original input feature vector, and $\phi(x)$ its feature-attribution vector. We define $x_k^+$ as the input where only the top-$k\%$ most important features (according to $\phi(x)$) are retained, and $x_k^-$ as the input where these top-$k\%$ features are masked. *Fidelity_* measures signal preservation when retaining top-$k\%$ features, computed as $\mathbb{E}[f(x_k^+) - f(x)]$, while *Fidelity_+* captures the effect of removing these features, computed as $\mathbb{E}[f(x_k^-) - f(x)]$. To assess stability, we employ Robustness to Feature Noise, where Gaussian noise $\epsilon \sim \mathcal{N}(0, 1)$ is added to obtain $x' = x + 0.05 * \sigma * \epsilon$, and robustness is quantified as $1 - \|\phi(x) - \phi(x')\|/\|\phi(x)\|$, where $\sigma$ denotes the standard deviation of the original features. Higher values indicate explanations that are both faithful and stable under perturbations.

**Datasets.** We evaluate on five widely used benchmark datasets for node classification: Amazon-Computers (D1) and Amazon-Photo (D2) from Shchur et al. (2018), CiteSeer (D3), PubMed (D4) and Cora (D5) from Yang et al. (2016). These datasets span diverse graph structures and feature

Table 2: Comparison of Fidelity$_+$ and Fidelity$_-$ scores for GNN-LRP, GOAT, IG, GradCAM, LIME, Random, and our self+2hop method across Amazon-Computers (D1), Amazon-Photo (D2), Citeseer (D3), PubMed (D4), and Cora (D5). Values are averaged over 100 random nodes, using $k = 5\%$ of features masked.

|  | Data | GOAT | **Ours** | GNN-LRP | IG | GradCAM | LIME | Random |
|---|---|---|---|---|---|---|---|---|
| Fidelity$_+$ | D1 | 0.0352 | **0.0353** | 0.0288 | 0.0164 | 0.0201 | 0.0070 | 0.0013 |
|  | D2 | 0.0340 | **0.0342** | 0.0227 | 0.0164 | 0.0024 | 0.0114 | 0.0009 |
|  | D3 | 0.1523 | **0.1514** | 0.1513 | 0.1239 | 0.1114 | 0.0281 | 0.0055 |
|  | D4 | 0.0389 | **0.0384** | 0.0313 | 0.0196 | 0.0215 | 0.0010 | 0.0010 |
|  | D5 | 0.1808 | **0.1789** | 0.1787 | 0.1229 | 0.1073 | 0.0267 | 0.0054 |
| Fidelity$_-$ | D1 | -0.0001 | **0.0000** | 0.0000 | -0.0002 | 0.0002 | 0.0004 | 0.0013 |
|  | D2 | -0.0001 | **0.0000** | 0.0000 | 0.0003 | 0.0046 | 0.0001 | 0.0010 |
|  | D3 | 0.0000 | **0.0000** | 0.0000 | 0 | 0.0002 | 0.0063 | 0.0063 |
|  | D4 | 0.0000 | **0.0000** | 0.0000 | -0.0003 | 0.0005 | 0.0011 | 0.0021 |
|  | D5 | 0.0000 | **0.0000** | 0.0000 | -0.0001 | 0.0006 | 0.0046 | 0.0033 |

distributions, making them standard for GNN explainability research. Dataset statistics (nodes, edges, feature dimensions, and class distributions) are summarized in Table 4.

## 4.2 RESULTS

Table 2 reports Fidelity$_+$ and Fidelity$_-$ scores across all datasets. Fidelity$_+$ measures the difference in prediction when the top-$5\%$ important features are removed compared to the original input. Larger values indicate that the retained features capture most of the predictive signal, while Fidelity$_-$ measures the difference when only the top-$5\%$ important features are kept. Here, smaller values are better, as they indicate that masking less relevant features does not significantly distort predictions. In summary, our method achieves state-of-the-art Fidelity$_+$, closely matching or surpassing GOAt on most of benchmarks and outperforming gradient-based and perturbation-based methods, demonstrating that the proposed decomposition effectively identifies features critical for preserving model confidence. Moreover, our method achieved near-zero Fidelity$_-$ values, confirming that the decomposition produces clean and stable feature rankings.

Robustness results, summarized in Table 5 (Appendix), show that our method outperforms IG, GNN-LRP, and LIME. The slightly lower robustness is expected because the proposed method operates directly in the input feature space, where additive noise proportionally perturbs the importance magnitudes. Nevertheless, robustness values remain high (0.78–0.98), demonstrating that the method is relatively stable. Finally, Table 6 (Appendix) compares computational efficiency. Our method achieves a favorable trade-off between efficiency and faithfulness.

We also investigate feature signal recovery under feature noise by augmenting Cora's 1,433 features with 287 Bernoulli noise columns (p=0.013), comparing our method against established baselines. A 2-layer GCN (details in Section A.5) is trained, and 100 test nodes are sampled. Each explainer ranks features; we measure the number of noisy features among the Top-10 and report runtime statistics. Results (Appendix 8) show our method consistently selects fewer noisy features, with PCA-based aggregation offering slight gains over averaging—indicating improved stability and signal recovery.

## 5 DISCUSSION

### 5.1 LIMITATIONS AND SCOPE

Our decomposition framework attributes embedding values to input features by propagating contributions through the network's computational graph, assuming access to model internals (parameters, intermediate activations, and stored normalization statistics) and achieving exactness only for specific activation and normalization classes. For linear transformations and piecewise-linear activations such as ReLU (Nair & Hinton, 2010) and LeakyReLU (Maas et al., 2013), contributions admit closed-form propagation via sign/magnitude masks or slope-based scaling. Monotone acti-

vations with tractable inverses (e.g., ELU (Clevert et al., 2016), SELU (Klambauer et al., 2017), Softplus (Nair & Hinton, 2010)) are handled by exact inversion when numerically stable, or by local linearization using their derivatives for saturating nonlinearities such as Sigmoid and Tanh, which are invertible in principle, we employ clamping or derivative-weighted masks to mitigate numerical instability near saturation. Modern smooth activations (GELU (Hendrycks & Gimpel, 2017), Swish/SiLU (Ramachandran et al., 2017), Mish (Misra, 2019)) lack simple closed-form inverses and can be non-monotonic. However, Hendrycks & Gimpel (2017) provides the approximated form of GELU with $f(x) = 0.5x \left(1 + \tanh\left[\frac{\sqrt{2}}{\pi}(x + 0.044715x^3)\right]\right)$, which can be inverted using a branch-aware Newton–Raphson method on $g(x) = f(x) - y$ with close-form $f'(x)$. Alternatively, DeepLIFT (Shrikumar et al., 2017) or LRP (Bach et al., 2015) provide principled propagation without explicit inversion for non-monotonic cases. Normalization layers exhibit analogous behavior: BatchNorm (Ioffe & Szegedy, 2015) is invertible at inference given stored statistics as discussed in Lu et al. (2024), while LayerNorm (Ba et al., 2016) depends on per-sample moments and is treated via local linearization. Pooling and attention are decomposed by distributing relevance proportionally to aggregation weights or attention scores, preserving interpretability in graph-based architectures.

The proposed approach assumes access to model internals—such as weights, activations, and normalization statistics. In black-box settings (e.g., API-based inference), these details are unavailable, making exact decomposition infeasible. This is not unique to our approach: widely used population/global explainers require access to model internals (Ying et al., 2019; Baldassarre & Azizpour, 2019; Luo et al., 2020; Lu et al., 2024). If internal access is restricted, alternative approaches like model-agnostic methods, perturbation-based sensitivity analysis, or surrogate modeling can approximate interpretability, albeit with reduced faithfulness.

Above, we used PCAas a post-hoc visualization tool to aggregate the contribution matrix in order to show overall patterns across interpretations. While PCA can be effective for summarizing high-dimensional data, it introduces an additional layer of abstraction. However, we note that PCA is not part of the explanation mechanism itself—the raw contribution matrix remains available for direct analysis. Alternative aggregation strategies that preserve interpretability are important future work, such as TCAV introduced by Kim et al. (2018), which aligns latent directions with human-defined concepts, or disentangled representation learning (Piaggesi et al., 2025), which aims to produce dimensions with clearer semantic meaning.

## 5.2 FUTURE WORK

**Graph Attention Networks (GAT).** As a future direction, we aim to extend our path-based decomposition framework to attention-based architectures such as GAT, which often outperform non-attention GNNs (Veličković et al., 2018). While we provide a detailed decomposition for a simplified two-layer GAT in Appendix A.6, this is only a preliminary step. Generalizing to deeper GATs, heterogeneous attention mechanisms, and residual connections remains an open challenge that we plan to explore in future work.

**Graph Transformer (GT).** Recent advances in Graph Transformers (Yun et al., 2019) pose a challenge to our path-based formulation. By replacing sparse neighborhoods with fully-connected attention, GTs allow each node to attend to all others, which eliminates the locality-based propagation structure that our method exploits. To address this challenge, we need to determine how to define paths through a dense attention mechanism without facing combinatorial explosion. One promising direction is to track only the most influential connections, potentially through iterative pruning or hierarchical clustering of attention patterns. Second, GTs commonly use positional encodings like Laplacian eigenvectors or shortest path distances capture structure, but these encodings are processed jointly with node features through the same attention mechanism. Separating the positional and node-feature contributions is an open problem, but necessary to capture the impact of the node features on the network's decision process.

**Uncertainty Quantification.** Beyond extending the framework to other convolutional architectures, an important research direction is to incorporate confidence measures into explanations. Current attribution methods—including ours—assume deterministic faithfulness within activation regions, but do not quantify uncertainty under distribution shifts or adversarial perturbations. Fu-

ture work could explore probabilistic decomposition techniques to estimate explanation reliability, enabling practitioners to assess whether an attribution remains trustworthy when the input graph deviates from training distribution.

# 6  CONCLUSION

This paper introduced a feature-wise decomposition framework for interpreting graph neural network embeddings. By reformulating GNN layers as linear contribution operators, our approach provides explicit attributions across both self and neighbor pathways, while a PCA-based aggregation strategy mitigates correlation bias among embedding dimensions. Experiments on homogeneous (Cora) and heterogeneous (MovieLens) benchmarks demonstrate that our method delivers fine-grained, semantically aligned explanations of predictive embeddings. These results underscore the value of embedding decomposition for revealing how relational signals shape learned representations, thereby advancing transparency and accountability in GNN-driven decision-making. Looking ahead, this work opens promising directions for extending the framework to deeper architectures, temporal or dynamic graphs, and high-stakes domains where interpretability is critical.

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

# A  APPENDIX

## A.1  NOTATION

| Symbol | Definition |
|---|---|
| $G = (\mathcal{V}, \mathcal{E})$ | Input graph with node set $\mathcal{V}$ and edge set $\mathcal{E}$ |
| $\mathbf{x}_v \in \mathbb{R}^{F_0}$ | Input feature vector of node $v$ |
| $X \in \mathbb{R}^{|\mathcal{V}| \times F_0}$ | Matrix of all input features |
| $\mathbf{h}_v^{(\ell)} \in \mathbb{R}^{F_\ell}$ | Embedding of node $v$ at layer $\ell$ |
| $W^{(\ell)}$ | Learnable weight matrix at layer $\ell$ |
| $W_r^{(\ell)}$ | Relation/type-specific weight matrix |
| $\mathbf{b}^{(\ell)}$ | Bias vector at layer $\ell$ |
| $\sigma$ | Nonlinear activation (e.g., ReLU) |
| $D_v^{(\ell)}$ | Diagonal mask from activation of node $v$ at layer $\ell$ |
| $N_r(v)$ | Neighbors of $v$ under relation/type $r$ |
| $\tilde{A}$ | Normalized adjacency matrix used in GCN |
| $\mathbf{e}_p$ | $p$-th standard basis vector in $\mathbb{R}^{F_0}$, i.e., a column vector with 1 in position $p$ and 0 elsewhere |
| $\mathbf{C}_{w \to v}^{(\ell)}$ | Contribution matrix from features of node $w$ to embedding of node $v$ at layer $\ell$ |

Table 3: Notation used in the Methods section.

## A.2  PCA-BASED CONTRIBUTION AGGREGATION

A central difficulty in aggregating feature contributions across embeddings is that the learned embeddings are often correlated. Directly summing raw contribution vectors may therefore over-count redundant information. One possible remedy is whitening, which rescales contributions by the inverse square root of the embedding covariance matrix (Zuber & Strimmer, 2011). However, in practice the covariance matrix $\Sigma_h$ may be ill-conditioned, and computing $\Sigma_h^{-1/2}$ can lead to numerical instability due to very small eigenvalues.

We instead adopt a principal component analysis (PCA) approach (Jolliffe & Cadima, 2016). Let $\Sigma_h = U \Lambda U^\top$ denote the eigen decomposition of the embedding covariance, with eigenvectors $U$

and eigenvalues $\Lambda = \text{diag}(\lambda_1, \ldots, \lambda_{F_L})$. We transform each contribution vector $\mathbf{c}_{w,p,\to v} \in \mathbb{R}^{F_L}$ into the orthogonal PCA basis such that $\hat{\mathbf{c}}_{w,p,\to v} = \mathbf{c}_{w,p,\to v} U$. The coordinates of $\hat{\mathbf{c}}_{w,p,\to v}$ now represent the effect of feature $p$ of node $w$ on independent directions of variation in the embedding space. A simple PCA-based importance score is then

$$s_{w,p\to v}^{\text{PCA}} = \|\hat{\mathbf{c}}_{w,p,\to v}\|_2,$$

which measures the overall magnitude of influence across decorrelated components.

Alternatively, one can weight contributions by the fraction of variance explained by its principal component:

$$s_{w,p\to v}^{\text{PCA-var}} = \left( \sum_{k=1}^{F_L} \frac{\lambda_k}{\sum_j \lambda_j} \left( \hat{\mathbf{c}}_{w,p\to v}[k] \right)^2 \right)^{1/2}.$$

This PCA-based approach avoids the instability of inverting $\Sigma_h$ while still capturing feature contributions along independent directions of variation in the embedding space. In practice, we often truncate to the top $K$ principal components, which both reduces noise and highlights contributions to dominant modes of variation.

## A.3 DATASET DESCRIPTIONS AND EXPERIMENTAL SETUP

**Cora (GCN case study).** The Cora citation network (Sen et al., 2008) consists of 2,708 scientific publications categorized into seven research areas, connected by 5,429 citation links. Each node represents a paper, and its feature vector is a bag-of-words (BoW) representation over 1,433 unique terms from the papers. The prediction task is node classification: given the citation graph and node features, predict the research category of each paper. We adopt the standard train/validation split from Kipf & Welling (2017), with the remaining nodes reserved for testing. This dataset provides a benchmark for evaluating our method in a transductive, homogeneous, single-type graph setting, where GCN serves as a natural baseline. For the experimental setup, we train a two-layer GCN, where each layer outputs 16 hidden dimensions, followed by ReLU activation and dropout with a rate of 0.5. The final layer is a softmax classifier over seven publication categories. Training is performed using the Adam optimizer with a learning rate of 0.01, minimizing the cross-entropy loss on labeled nodes. We use 140 nodes for training, 500 for validation, and 2,068 for testing. Early stopping is applied based on validation accuracy with a patience of 10 epochs.

**MovieLens (HinSAGE case study).** The MovieLens dataset (Harper & Konstan, 2015) comprises user–movie interactions represented as a bipartite heterogeneous graph. We use the 100K subset, which contains 100,000 ratings from 943 users on 1,682 movies. Nodes correspond to users and movies, while edges denote rating interactions. Node features include auxiliary attributes such as movie genres and user profiles. Each edge is associated with an integer rating in the range [1,5]. We formulate the task as supervised link-attribute regression: given a user node, a movie node, and their attributes, the model predicts the rating on the corresponding edge. This setting evaluates our framework under an inductive, heterogeneous, edge-level prediction scenario. Our model adopts a one-layer HinSAGE architecture as suggested in Zhang & Chen (2020) with a hidden dimension of 16 and a mean aggregator. For each target node, the model samples neighborhoods of size 200 to compute node embeddings. Edge embeddings are constructed by concatenating the embeddings of user–movie pairs and passing them through a dense layer of size 16 with a linear activation, followed by a single linear output unit to produce a continuous rating prediction. The model is optimized using Adam with a learning rate of 0.01 and trained with mean squared error (MSE) as the objective. We allocate 60,000 edges for training, 10,000 for validation, and 30,000 for testing. Early stopping is applied based on validation mean absolute error (MAE) with a patience of 5 epochs.

## A.4 ADDITIONAL RESULTS

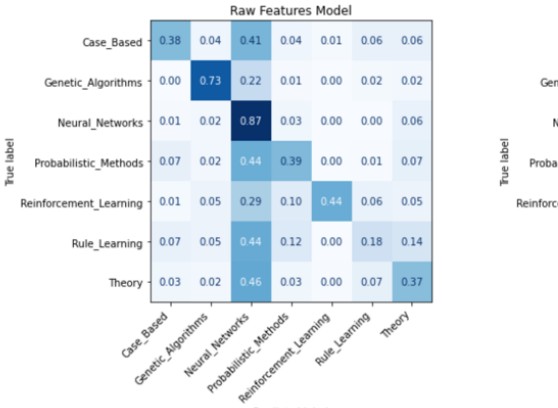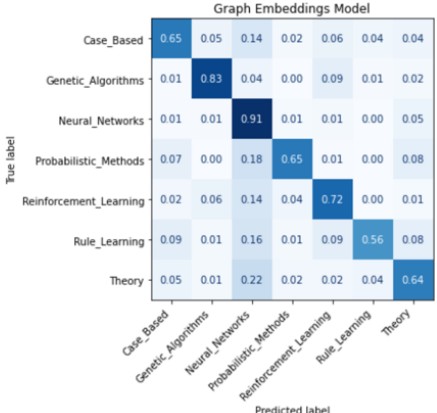

Figure 3: Confusion matrices of XGBoost trained with (a) raw bag-of-words features and (b) GCN-derived embeddings. The GCN embeddings produce a more diagonally dominant pattern and suppress structured off-diagonal blocks, indicating improved class separability and reduced systematic confusion among semantically related classes. Labels are ordered consistently across panels.

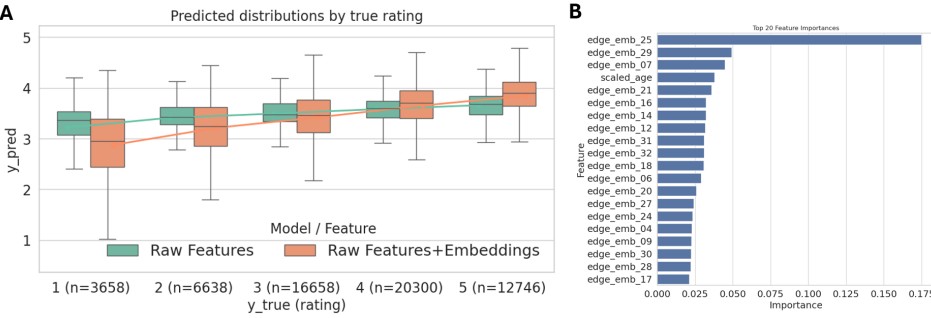

Figure 4: A. Model performance comparison for XGBoost regression on the MovieLens dataset using either raw features or raw + HinSAGE embeddings. Boxplots show predicted rating distributions (y-axis) grouped by true rating levels (x-axis). Quantitatively, adding embeddings decreases mean squared error (MSE) and mean absolute error (MAE) while increasing $R^2$, demonstrating that learned representations capture latent structure beyond raw features. B. Feature importance (measured by total gain) from the XGBoost regression model. Embedding dimensions dominate the top-ranked features (e.g., edge_emb_25, edge_emb_29, edge_emb_07), indicating that learned graph-based representations contribute more to predictive performance than raw user attributes.

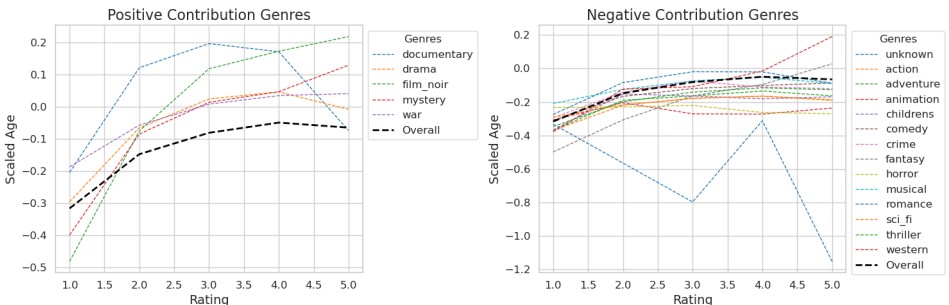

Figure 5: Average contributions of the scaled age attribute to the $25^{th}$ graph embedding across rating levels (x-axis) and aggregated by movie genres. The left panel shows genres where age exhibits positive contributions (e.g., documentary, drama, film noir, mystery, war), while the right panel shows genres with negative contributions (e.g., action, animation, comedy, horror). Each curve represents the marginal effect of scaled age on the model's output for a given genre, averaged over users and items. Positive-contribution genres generally show an increasing trend with rating level, indicating that older users are associated with higher ratings for these genres. Conversely, negative-contribution genres seem to exhibit decreasing or flat trends, suggesting that younger users tend to give higher ratings in these categories. The overall trend (black dashed line) summarizes the global effect across all genres.

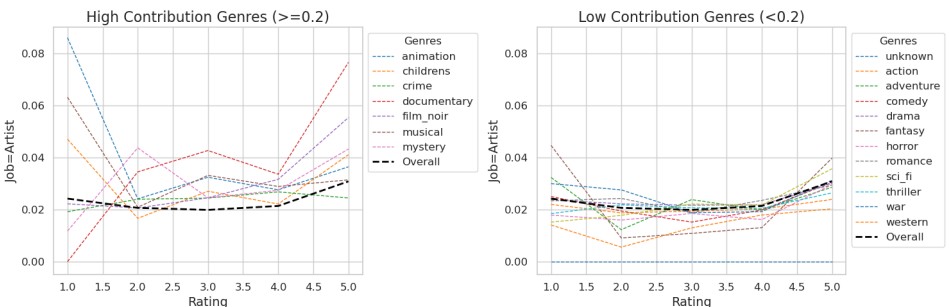

Figure 6: Average contributions of the Artist job feature to the $25^{th}$ graph embedding across rating levels (x-axis), aggregated by movie genres. The left panel shows genres with high contribution ($\geq 0.2$), including animation, children's, crime, documentary, film noir, musical, and mystery. The right panel shows genres with low contribution ($< 0.2$), such as action, adventure, comedy, drama, horror, and others. High-contribution genres tend to exhibit stronger positive sensitivity to higher ratings, whereas low-contribution genres remain relatively flat, indicating limited influence of the Artist feature in those categories. The black dashed line denotes the overall trend across all genres.

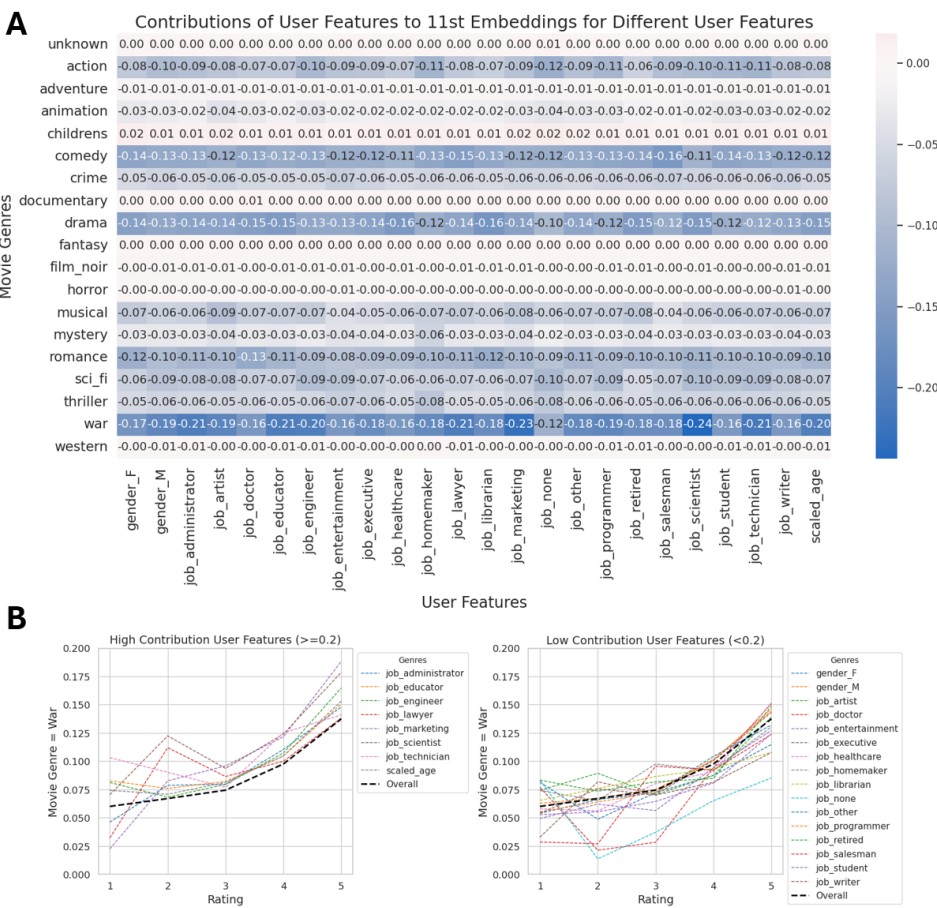

Figure 7: A. Analysis of the $11^{th}$ embedding dimension, showing aggregated contributions across movie genres (rows) and user attributes (columns). Each cell represents the marginal effect of a user feature on this embedding dimension for a given genre, with darker shades indicating stronger negative contributions. user attributes include gender, occupation categories, and scaled_age, which is treated as a binary feature here (results correspond to cases where scaled_age $> 0$). Analysis of the $11^{th}$ embedding dimension, showing aggregated contributions across movie genres (rows) and user attributes (columns). Each cell represents the marginal effect of a user feature on this embedding dimension for a given genre, with darker shades indicating stronger negative contributions. User attributes include gender, occupation categories, and scaled_age, which is treated as a binary feature here (results correspond to cases where scaled_age $> 0$). B. Visualization of average user feature contributions for the war genre across rating levels (x-axis). The left panel shows features with high aggregate contribution ($\geq 0.2$), primarily occupational attributes (e.g., administrator, educator, engineer, scientist) and scaled_age (binary, $> 0$). The right panel shows features with low aggregate contribution ($< 0.2$), including gender and less influential occupations. Each curve represents the marginal effect of a user feature on predicted ratings for war movies, averaged across users and items. The curves cluster closely, indicating that differences among these features are subtle and not strongly discriminative for this genre.

## A.5  COMPARATIVE EVALUATION ON NOISY CORA

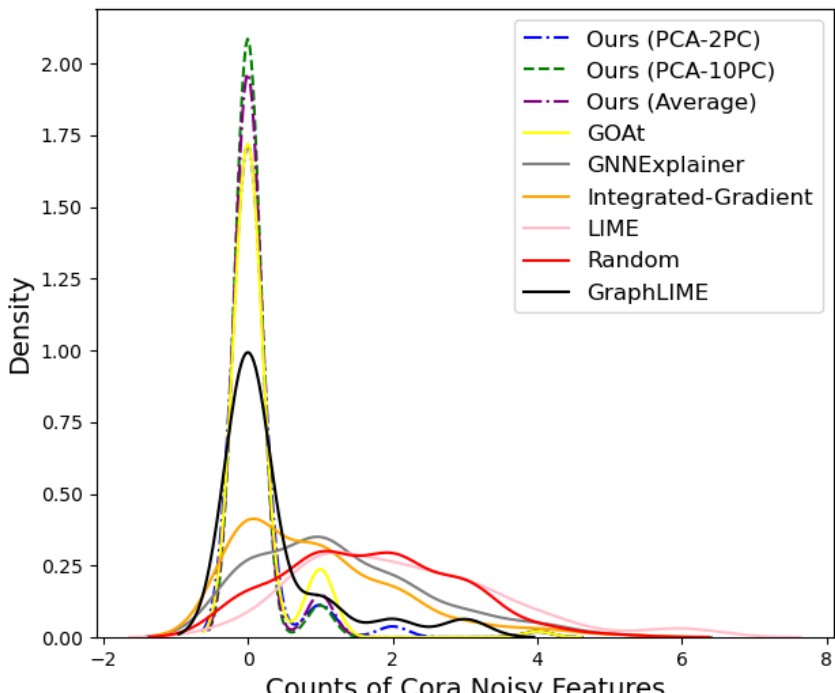

Figure 8: Frequency distributions of noisy features across different explanation methods using a GCN model on the Cora dataset.

We compare our decomposition based approach with established baselines: GOAt (Lu et al., 2024), GNNExplainer (Ying et al., 2019),Integrated Gradients (Sundararajan et al., 2017a),LIME (Ribeiro et al., 2016), GraphLIME (Huang et al., 2022), and a Random explainer. Our method inverts layerwise embeddings back to raw features and aggregates contributions either by simple averaging across embedding dimensions or via a PCA based decorrelation. We evaluate how effectively competing explanation methods suppress uninformative (noise) node features on the Cora dataset. Following prior work(Huang et al., 2022; Duval & Malliaros, 2021), we augment the original 1,433 bag of words features with 287 additional Bernoulli (p=0.013) noise columns, which have a similar distribution as existing features. We train a 2 layer GCN with the same settings as in Section 3.1. We sample 100 test nodes; each explainer produces a feature importance vector, and we count how many of the Top 10 ranked features are the added noisy features. We report the mean and standard deviation of the runtime per node across these 100 nodes.

Figure 8 shows that, across the sampled nodes, our method selects the fewest noisy features in general, indicating higher attribution precision. For the PCA-based aggregation, using more principal components (e.g., 10 PCs) seem to achieve a small but consistent gain over fewer PCs (e.g., 2 PCs) and over simple averaging—suggesting that PCA-based aggregation helps to improve stability and signal recovery. Overall, the decomposition framework attains a favorable fidelity–efficiency trade off compared to existing baselines.

Table 4: Statistics of datasets used in experiments. Each dataset is labeled as D1–D5 for reference.

| ID | Dataset | Graphs | Nodes | Edges | Features | Classes |
|----|---------|--------|-------|-------|----------|---------|
| D1 | Computers | 1 | 13,752 | 491,722 | 767 | 10 |
| D2 | Photo | 1 | 7,650 | 238,162 | 745 | 8 |
| D3 | CiteSeer | 1 | 3,327 | 9,104 | 3,703 | 6 |
| D4 | PubMed | 1 | 19,717 | 88,648 | 500 | 3 |
| D5 | Cora | 1 | 2,708 | 10,556 | 1,433 | 7 |

Table 5: Robustness$_\text{feat}$ for GNN-LRP, GOAT, IG, GradCAM, LIME, Random, and our method across Amazon-Computers (D1), Amazon-Photo (D2), Citeseer (D3), PubMed (D4), and Cora (D5). Values are averaged over 100 random nodes, using the top $5\%$ important features.

| Method | D1 | D2 | D3 | D4 | D5 |
|--------|------|------|------|------|------|
| GradCAM | 0.9987 | 0.9949 | 0.9900 | 0.9908 | 0.9866 |
| GOAT | 0.9934 | 0.9927 | 0.8350 | 0.9852 | 0.9530 |
| Ours | 0.9774 | 0.9800 | 0.7820 | 0.9596 | 0.8576 |
| IG | 0.9627 | 0.9545 | 0.2728 | 0.9791 | 0.2493 |
| GNN-LRP | 0.6553 | 0.6588 | 0.1929 | 0.5810 | 0.1267 |
| Random | 0.5089 | 0.5087 | 0.5119 | 0.5229 | 0.5149 |
| LIME | 0.3879 | 0.3962 | 0 | 0.0270 | 0.0072 |

Table 6: Per-node explanation runtime (seconds) for all methods across Amazon-Computers (D1), Amazon-Photo (D2), Citeseer (D3), PubMed (D4), and Cora (D5). Reported values are averaged over 100 randomly selected target nodes.

| Method | D1 | D2 | D3 | D4 | D5 |
|--------|------|------|------|------|------|
| Random | 0.040 | 0.023 | 0.047 | 0.038 | 0.017 |
| GradCAM | 0.132 | 0.071 | 0.058 | 0.065 | 0.021 |
| GNN-LRP | 0.327 | 0.535 | 0.308 | 0.535 | 0.269 |
| Ours | 45.782 | 13.857 | 0.193 | 0.749 | 0.486 |
| GOAT | 220.752 | 74.038 | 27.527 | 437.584 | 14.792 |
| LIME | 495.143 | 161.958 | 92.927 | 118.167 | 43.122 |
| IG | 611.970 | 252.356 | 194.829 | 224.865 | 56.606 |

## A.6 DECOMPOSITION FOR TWO-LAYER GAT

For layer $\ell$ and head $k$, with input features $\mathbf{h}_j^{(\ell)} \in \mathbb{R}^{F_\ell}$,

$$\mathbf{z}_j^{(\ell,k)} = \mathbf{W}^{(\ell,k)}\mathbf{h}_j^{(\ell)} \in \mathbb{R}^{F'_\ell},$$

$$e_{ij}^{(\ell,k)} = \text{LeakyReLU}\left(\mathbf{a}^{(\ell,k)\top}\left[\mathbf{z}_i^{(\ell,k)} \,\|\, \mathbf{z}_j^{(\ell,k)}\right]\right),$$

$$\alpha_{ij}^{(\ell,k)} = \frac{\exp(e_{ij}^{(\ell,k)})}{\sum_{t\in\mathcal{N}(i)}\exp(e_{it}^{(\ell,k)})},$$

$$\mathbf{u}_i^{(\ell,k)} = \sum_{j\in\mathcal{N}(i)} \alpha_{ij}^{(\ell,k)}\mathbf{z}_j^{(\ell,k)}, \qquad \mathbf{h}_i^{(\ell+1)} = \phi_\ell\big(\text{AGG}_k(\mathbf{u}_i^{(\ell,k)})\big),$$

where $\mathbf{W}$ denote learnable weights, $\mathbf{a}^{(\ell,k)\top}$ is a single-layer feedforward neural network and $\text{AGG}_k$ is concatenation in hidden layers and averaging/sum in the final layer. For node $i$, the contribution from neighbor $j$'s feature $f$ to output coordinate $r$ in layer $\ell$, head $k$, *before* the activation function is $C_{i\leftarrow j,\, f\rightarrow r}^{(\ell,k)} = \alpha_{ij}^{(\ell,k)} W_{r,f}^{(\ell,k)} h_{j,f}^{(\ell)}$. Summing over $j$ yields per-feature importances $C_i^{(\ell,k),\text{feat}} \in \mathbb{R}^{F_\ell \times F'_\ell}$; summing over $f$ yields per-neighbor importances $C_i^{(\ell,k),\text{neigh}} \in \mathbb{R}^{N\times F'_\ell}$. We pass the pre-activations through a local linear gate $G_i^{(\ell,k)} = \text{diag}\big(\phi'_\ell(\mathbf{u}_i^{(\ell,k)})\big)$ to account for the nonlinearity at the operating point, where $\phi'_\ell$ is the derivative of the activation function used in layer $\ell$.

For example, for two-layer GAT, let layer 1 have $K_1$ heads of width $F'_1$ (concatenated width $K_1 F'_1$), and layer 2 have a single head of width $F'_2$ (no concatenation). Denote $\mathbf{W}^{(1,k)} \in \mathbb{R}^{F'_1 \times F_0}$, $\mathbf{W}^{(2)} \in \mathbb{R}^{F'_2 \times (K_1 F'_1)}$, and the layer-1 local gate entries by $G_{j,q}^{(1,k)} = \phi'_1\big(u_{j,q}^{(1,k)}\big)$. Then the contribution from source node $u$'s input feature $f$ to the *layer-2 pre-activation* coordinate $r$ at target node $i$ is

$$C_i[u, f \rightarrow r] = \sum_{j\in\mathcal{N}(i)} \alpha_{ij}^{(2)} \sum_{k=1}^{K_1}\sum_{q=1}^{F'_1} W_{r,\,(k-1)F'_1+q}^{(2)}\, G_{j,q}^{(1,k)}\, \alpha_{ju}^{(1,k)}\, W_{q,f}^{(1,k)}\, h_{u,f}^{(0)}.$$

If the final embedding is defined *post-activation* for layer 2, multiply the right-hand side by the layer-2 gate $G_{i,r}^{(2)} = \phi'_2\big(u_{i,r}^{(2)}\big)$.

## A.7 USE OF LARGE LANGUAGE MODELS

We adopt a large language model (Copilot, GPT-5) to help polish the writing of the manuscript such as improving grammar and readability. All content was verified and revised by the authors.

