# OpenReview forum: "Interpretable Graph Embeddings: Feature-Level Decomposition for Trustworthy Graph Neural Networks"
_ICLR.cc/2026/Conference — Submitted to ICLR 2026_

### Official Review · Reviewer_7hVM · 2025-10-28

**Soundness:** 1
**Presentation:** 1
**Contribution:** 1
**Rating:** 2
**Confidence:** 4

**Summary:**

This paper proposes a method to decompose GNN embeddings. The core idea is that for each node (or edge) embedding dimension, trace exactly which raw features from which nodes contributed to it, and by how much. The authors argue that this matters in practice because graph embeddings are often exported as engineered features into downstream models. Then the paper walks through this idea for a 2-layer GCN on Cora and for a 1-layer HinSAGE / GraphSAGE-style model on MovieLens. After reading the main text and appendix, I lean toward **rejection rating since the empirical evidence is mostly case studies with limited quantitative rigor, some important details are under-specified or internally inconsistent, the writing and presentation quality are poor, and the required ICLR submission compliance items (ethics statement, reproducibility statement) are also missing.**

**Strengths:**

1. Clear practical motivation. The paper targets a nice point: people deploy GNNs to generate embeddings, then feed those embeddings into non-GNN models, and regulators ask what those embedding dimensions mean.
2. Concrete layer-wise accounting. For simple message passing with ReLU, the paper gives an explicit linear expansion that attributes each embedding coordinate of node $v$ to specific source features along $w \to u \to v$ paths, and then repeats the exercise on a heterogeneous user–movie graph to separate self-node vs. neighbor-type contributions. I think this is technically transparent and easy to follow for shallow GCN layers.

**Weaknesses:**

1. **Insufficient quantitative evidence and unclear evaluation protocol.**
   - The Cora study reports that replacing bag-of-words with learned GCN embeddings boosts downstream XGBoost performance (accuracy 0.57→0.76, weighted F1 0.56→0.75), and the MovieLens study reports that adding HinSAGE-derived user/movie embeddings to a regressor improves MSE, MAE, and $R^2$.     However, the paper does not fully spell out how training vs. testing separation is enforced when those downstream models consume embeddings. For MovieLens in particular, it is not explained whether held-out user–movie rating edges are completely excluded from the message-passing graph during training or whether information can leak in through sampled neighborhoods.   This matters for how strong the reported gains really are.
   - Also, despite repeatedly calling the method “exact,” there is no sanity check that the summed per-feature attributions numerically reconstruct the original embedding vector with near-zero error. Finally, the “noisy Cora” benchmark is only described qualitatively. I do not see concrete numbers, baseline hyperparameters, so it is hard to verify the claims of best noise suppression.

2. **Over-claiming determinism and inconsistency.**
   - The paper’s pitch is that attribution is deterministic, closed-form, and faithful. For a 2-layer GCN with fixed ReLU masks, that is largely true.   But later sections acknowledge that for smoother activations (GELU, Swish, Mish), LayerNorm / BatchNorm, attention, or pooling, the method needs approximations, local linearization, or heuristic redistribution of contribution.   Likewise, HinSAGE relies on stochastic neighborhood sampling, and the text itself says explanations are “conditional on the sampled computation graph,” suggesting that multiple samples must be averaged to get stable attributions. That weakens the claim of strict determinism.
   - In addition, in the MovieLens case study the narrative sometimes refers to “dimension 25,” “dimension 11,” and four disjoint 8-D blocks (user self, user’s neighbor movies, movie self, movie’s neighbor users), but Appendix A.3 then says the concatenated user/movie embeddings are passed through an extra dense layer before regression.     It is unclear which vector (pre-MLP concat vs post-MLP hidden) is actually being interpreted when making age/genre claims. This ambiguity undercuts the “faithful, auditable story” the paper promises.

3. **Reproducibility and reporting gaps.**
   - The paper does not provide ICLR required reproducibility statement and any other related stuff like code, anonymized repo, or even pseudocode for the attribution backprop and the PCA aggregation step that collapses 16 embedding dims into 5 “principal explanation axes.”   It also does not clearly define which PCA-based score ($s^{\text{PCA}}$ or $s^{\text{PCA-var}}$ in the appendix) is used to generate each figure. These missing details reduce the soundness of the experiment results and make it hard to verify or reuse the method.

4. **Writing and presentation issues.**
   The writing quality is poor. And there are repeated sentences, spacing/typo issues (e.g., “interprete” in line 255). The paper pasted an image as a table in Fig. 3 (Actually, this is greatly discouraged from the ICLR template). The future work section in 4.2 seems not give any insightful analysis and just lists GAT and GT (especially on GT, the whole paragraph is just a common intro and conveys nothing related to this paper)

5. **Missing required ICLR statements.**
   The submission lacks an Ethics Statement and a Reproducibility Statement, and it does not include anonymous code or supplementary artifacts.

**Questions:**

1. Downstream usage protocol. When you train the external XGBoost (for Cora classification and for MovieLens rating regression), how exactly do you split train vs. test so that no test embeddings or edges leak into training, even indirectly through message passing or neighbor sampling? Please spell out that protocol and confirm that the reported accuracy / F1 / MSE / MAE / $R^2$ numbers are on held-out data only.

2. Faithfulness sanity check. Can you report the $\ell_2$ reconstruction error between a learned embedding vector $h_v$ and the sum of all propagated per-feature contributions plus bias terms? This would give direct numerical evidence for the claimed “exact” decomposition.

3. Determinism vs. sampling. For HinSAGE, you mention that explanations are conditional on the sampled computation graph and could be averaged over multiple samples. How much variance do you actually observe across samples in practice, and is that variance acceptable for audit scenarios?

4. PCA aggregation. In Figures where you say “dimension 11 drives younger users to like sci-fi,” which representation are we talking about: the raw concatenated [user emb ∥ movie emb] vector, or the 16-D dense layer after concatenation?

5. Baselines in the noisy-feature stress test. For GNNExplainer, Integrated Gradients, GraphLIME, GOAt, etc., what hyperparameters and runtime settings did you use, and can you share the numeric counts of noisy features recovered in Top-10 plus mean ± std runtime? Without numbers, it is hard to judge the “we are both cleaner and faster” claim.

**Details Of Ethics Concerns:**

This paper lacks the required Ethics Statement in the submission. Personally, I think no ethics review is needed, yet the MovieLens case explicitly analyzes demographic attributes such as age and gender as explanatory drivers for recommendation-like predictions, which may raise related concerns, so giving the required Ethics Statement in the paper is necessary.

---

> ### Author Response · Authors · 2025-11-21
>
> **W1**: We thank the reviewer for these detailed and important observations. For Cora, embeddings were computed using the train/validation/test node split suggested in Yang et al. (2016) and Shchur et al. (2018), who used 20 samples per class (thus 140 in total for 7 classes) for training. Downstream XGBoost models were trained only on embeddings from the training set and evaluated on the test set. For MovieLens, we use 60% of edges for training, 10% early stop and the rest for testing as described in the Appendix. held-out user–movie rating edges were completely excluded from the message-passing graph during training. Neighborhood sampling was restricted to observed edges only, ensuring no leakage from test interactions.We conducted a sanity check experiment, and the sum of per-feature contributions reconstructs the original embedding vector with zero error.
>
> **W2**:
> We thank the reviewer for the insightful feedback. Our goal is to faithfully interpret the representations learned by the GNN, rather than characteristics of the training samples. For HinSAGE, the computation graph depends on sampled neighborhoods, which introduces variance across runs. This is not a limitation of our method but an inherent property of inductive sampling-based GNNs.
> We thank the reviewer for pointing this out. In the current HinSAGE configuration, we employed 2-hop neighbors. Figure 2 illustrates how each embedding is constructed. For example, Dimension 11 represents embeddings derived from the selected user’s neighboring movie nodes, multiplied by the learned weights and bias (i.e., post-MLP), resulting in the final embedding. Similarly, Dimension 25 corresponds to embeddings generated from the selected movie’s neighboring user nodes using the same process. For reference, we have attached the model summary, where the “post-MLP” operations occur in mean_hin_aggregator and mean_hin_aggregator_1. We would like to attach the model summary as well:
>
>
> __________________________________________________________________________________________________
> **Layer (type)**                   | **Output Shape**       | **Param #** | **Connected to**
> -----------------------------------|------------------------|-------------|---------------------------------
> input_3 (InputLayer)               | (None, 200, 19)       | 0           | []
> input_4 (InputLayer)               | (None, 200, 24)       | 0           | []
> input_1 (InputLayer)               | (None, 1, 24)         | 0           | []
> reshape (Reshape)                  | (None, 1, 200, 19)    | 0           | ['input_3[0][0]']
> input_2 (InputLayer)               | (None, 1, 19)         | 0           | []
> reshape_1 (Reshape)                | (None, 1, 200, 24)    | 0           | ['input_4[0][0]']
> dropout_1 (Dropout)                | (None, 1, 24)         | 0           | ['input_1[0][0]']
> dropout (Dropout)                  | (None, 1, 200, 19)    | 0           | ['reshape[0][0]']
> dropout_3 (Dropout)                | (None, 1, 19)         | 0           | ['input_2[0][0]']
> dropout_2 (Dropout)                | (None, 1, 200, 24)    | 0           | ['reshape_1[0][0]']
> mean_hin_aggregator (MeanHinAggregator) | (None, 1, 16)    | 360         | ['dropout_1[0][0]', 'dropout[0][0]']
> mean_hin_aggregator_1 (MeanHinAggregator) | (None, 1, 16) | 360         | ['dropout_3[0][0]', 'dropout_2[0][0]']
> reshape_2 (Reshape)                | (None, 16)            | 0           | ['mean_hin_aggregator[0][0]']
> reshape_3 (Reshape)                | (None, 16)            | 0           | ['mean_hin_aggregator_1[0][0]']
> lambda (Lambda)                    | (None, 16)            | 0           | ['reshape_2[0][0]', 'reshape_3[0][0]']
> link_embedding (LinkEmbedding)     | (None, 32)            | 0           | ['lambda[0][0]', 'lambda[1][0]']
> dense (Dense)                      | (None, 1)             | 33          | ['link_embedding[0][0]']
> reshape_4 (Reshape)                | (None, 1)             | 0           | ['dense[0][0]']
> __________________________________________________________________________________________________
> **Total params:** 753
> **Trainable params:** 753
> **Non-trainable params:** 0
> __________________________________________________________________________________________________
>
>
> Reference:
> Z. Yang, W. W. Cohen, and R. Salakhutdinov. Revisiting semi-supervised learning with graph embeddings. ICML, 2016
> Oleksandr Shchur, Maximilian Mumme, Aleksandar Bojchevski, and Stephan Günnemann. Pitfalls of Graph Neural Network Evaluation. In NIPS workshop, 2018

---

> ### Author Response · Authors · 2025-11-21
>
> **W3**: We thank the reviewer for highlighting this important point. We have provided code for the reproduction. We would like to provide more information here to illustrate both scores:
>
> - Step1: Run PCA on all transaction embeddings to obtain eigenvectors $v_1, \dots, v_d \in \mathbb{R}^d$ and corresponding eigenvalues $\lambda_1, \dots, \lambda_d$.
>
> - Step 2: Project each feature’s contribution vector onto the principal components, such that  $c_f^* = V^\top c_f$, where $c_{(f,j)}^* = v_j^\top c_f$ represents the contribution of feature falong principal component j.
>
> - Step 3: Compute aggregated scores as $s_{pca} = \sum_{j=1}^d \big( C[f,:] v_j \big)^2$, or leverage eigenvalues to get $s_{pca-var}$.
>
> **W4** : We thank the reviewer for bringing up these concerns. In the revised manuscript, we carefully proofread for language and formatting. We replaced image-based table with proper LaTeX table. We also modified Section 4.2 (now 5.2) in terms of names and clarifying the extensions to this work related to GT. For example, in the revised manuscript, we added:
>
> "Recent advances in Graph Transformers pose a challenge to our path-based formulation. By replacing sparse neighborhoods with fully-connected attention, GTs allow each node to attend to all others, which eliminates the locality-based propagation structure that our method exploits. To address this challenge, we need to determine how to define paths through a dense attention mechanism without facing combinatorial explosion. One promising direction is to track only the most influential connections, potentially through iterative pruning or hierarchical clustering of attention patterns. Second, GTs commonly use positional encodings like  Laplacian eigenvectors or shortest path distances capture structure, but these encodings are processed jointly with node features through the same attention mechanism.  Separating the positional and node-feature contributions is an open problem, but necessary to capture the impact of the node features on the network's decision process."
>
> **W5**: We thank the reviewer’s comment. We have provided the requested code. Regarding the Ethics Statement, according to the Author Guidelines, submissions without potential ethical concerns are not required to include such a statement.
>
> **Question 1**: We adopted the same settings used for GNN training when training the XGBoost model. For data splitting, nodes were randomly split in the Cora dataset, and edges were randomly split in the MovieLens dataset. Importantly, we did not include validation or test datasets in the training process for either GNN or XGBoost. After reviewing our experiments, we confirmed that all reported metrics—accuracy, F1, MSE, and MAE—were computed exclusively on held-out data.
> Please see our response in W1.
>
> **Question 2**: We thank the reviewer for this valuable comment. Please see our response in W1.
>
> **Question 3**: We thank the reviewer for raising this important point about determinism and sampling variance in HinSAGE. While we agree that checking variance is insightful and important step to make sure the stability of the explanation, we believe the overall pattern is the key to understand in practice. For example, in the application of fraud detection the key question is whether explanations remain stable enough to identify suspicious patterns (e.g., whether neighboring information from one specific relational entities dominates or not). we conducted the robustness test and compared with other approaches, the results showed that our method performed relatively stable. Results attached here: We also checked the results in this example and didn’t find much differences.
>
> Robustness:
>  | Method    | D1      | D2      | D3      | D4      | D5      |
> |:----------|---------|---------|---------|---------|---------|
> | GradCAM   | 0.9987  | 0.9949  | 0.9900  | 0.9908  | 0.9866  |
> | GOAT      | 0.9934  | 0.9927  | 0.8350  | 0.9852  | 0.9530  |
> | Ours      | 0.9774  | 0.9800  | 0.7820  | 0.9596  | 0.8576  |
> | IG        | 0.9627  | 0.9545  | 0.2728  | 0.9791  | 0.2493  |
> | GNN-LRP   | 0.6553  | 0.6588  | 0.1929  | 0.5810  | 0.1267  |
> | Random    | 0.5089  | 0.5087  | 0.5119  | 0.5229  | 0.5149  |
> | LIME      | 0.3879  | 0.3962  | 0       | 0.0270  | 0.0072  |
>
> **Question 4**: We thank the review for raising this question. Please our reply in W2 above.

---

> ### Author Response · Authors · 2025-11-21
>
> **Question 5**:
>
> In this experiment, we used torch_geometric.explain.Explainer with the following configurations:
>  - Integrated Gradients (IG): implemented via CaptumExplainer(IntegratedGradients).
>  - GNNExplainer: configured with algorithm = GNNExplainer and trained for 50 epochs.
>  - GraphLIME: based on the authors’ implementation with default hyperparameters (hop = 2, rho = 0.1).
>  - Other competing methods did not require explicit hyperparameter tuning.
>
> We would also like to share the numeric counts of noisy features:
>
> | # Noisy Features | Ours (PCA-2PC) | Ours (PCA-10PC) | Ours (MEAN) | GOAt | GNNExplainer | IG  | LIME | Random | GraphLME |
> |-------------------|----------------|-----------------|-------------|------|--------------|-----|------|--------|----------|
> | 0                | 92             | 93              | 91          | 87   | 28           | 43  | 6    | 16     | 79       |
> | 1                | 8              | 5               | 6           | 12   | 36           | 31  | 33   | 30     | 11       |
> | 2                | 0              | 0               | 2           | 0    | 21           | 17  | 26   | 29     | 5        |
> | 3                | 0              | 1               | 0           | 0    | 9            | 5   | 21   | 20     | 5        |
> | 4                | 0              | 0               | 1           | 1    | 5            | 3   | 9    | 4      | 0        |
> | 5                | 0              | 1               | 0           | 0    | 1            | 1   | 1    | 1      | 0        |
> | 6                | 0              | 0               | 0           | 0    | 0            | 0   | 4    | 0      | 0        |
> | 7                | 0              | 0               | 0           | 0    | 0            | 0   | 0    | 0      | 0        |

---

### Official Review · Reviewer_mPHL · 2025-10-30

**Soundness:** 2
**Presentation:** 2
**Contribution:** 2
**Rating:** 2
**Confidence:** 4

**Summary:**

This paper introduces a novel framework for interpreting graph neural networks (GNNs) by focusing on the decomposition of embeddings at the feature level. The authors propose an technique for GNN layers, transforming each layer into a contribution pathway, which allows fine-grained attribution across heterogeneous feature streams. The framework is demonstrated on two GNN architectures: Graph Convolutional Networks (GCN) and Heterogeneous GraphSAGE (HinSAGE), with experiments conducted on Cora and MovieLens datasets.

**Strengths:**

1.	The paper is well-written and clearly organized.

**Weaknesses:**

1.	What is the difference and relationship between the method proposed by the authors and the GNN-LRP [1] method? Is it a variant of the GNN-LRP method or a new approach for contribution allocation?
2.	The authors should compare their method with more path-based explanation methods for GNNs, such as GNN-LRP [1], FlowX [2], and AxiomPath-Convex [3], as the method proposed by authors also calculates contribution values ​​based on paths.
3.	The authors should test their method on more datasets, as the current number of datasets is too limited.

[1] Schnake T, Eberle O, Lederer J, et al. Higher-order explanations of graph neural networks via relevant walks[J]. IEEE transactions on pattern analysis and machine intelligence, 2021, 44(11): 7581-7596.

[2] Gui S, Yuan H, Wang J, et al. Flowx: Towards explainable graph neural networks via message flows[J]. IEEE Transactions on Pattern Analysis and Machine Intelligence, 2023, 46(7): 4567-4578.

[3] Liu Y, Zhang X, Xie S. A differential geometric view and explainability of gnn on evolving graphs[J]. arXiv preprint arXiv:2403.06425, 2024.

**Questions:**

See the weaknesses.

---

> ### Author Response · Authors · 2025-11-21
>
> **W1**: We thank the reviewer for raising this important question regarding the relationship between our method and GNN-LRP. Our method is not a variant of the GNN-LRP. GNN-LRP focuses on relevance propagation along graph walks, assigning relevance scores to nodes and edges based on higher-order paths. It explains predictions by tracing influential walks in the graph structure, while our method provides a feature-level decomposition of node embeddings, attributing contributions from input features to each embedding dimension. GNN-LRP explains why a node is classified a certain way, while our method explains how input features shape the learned representation.
>
> **W2**: We thank the reviewer for this constructive suggestion regarding comparisons with path-based explanation methods. We have included GNN-LRP as one of the competitors in our newly added quantitative experiments. These newly added experiments use widely adopted metrics (Fidelity+, Fidelity-, Robustness) across five benchmark datasets. The results are summarized in the revised manuscript.
>
> | Metric       | Data | GOAT    | Ours    | GNN-LRP | IG      | GradCAM | LIME    | Random  |
> |:------------|------|---------|---------|---------|---------|---------|---------|---------|
> | Fidelity₊   | D1   | 0.0352  | **0.0353** | 0.0288  | 0.0164  | 0.0201  | 0.0070  | 0.0013  |
> |             | D2   | 0.0340  | **0.0342** | 0.0227  | 0.0164  | 0.0024  | 0.0114  | 0.0009  |
> |             | D3   | 0.1523  | **0.1514** | 0.1513  | 0.1239  | 0.1114  | 0.0281  | 0.0055  |
> |             | D4   | 0.0389  | **0.0384** | 0.0313  | 0.0196  | 0.0215  | 0.0010  | 0.0010  |
> |             | D5   | 0.1808  | **0.1789** | 0.1787  | 0.1229  | 0.1073  | 0.0267  | 0.0054  |
> | Fidelity₋   | D1   | -0.0001 | **0.0000** | 0.0000  | -0.0002 | 0.0002  | 0.0004  | 0.0013  |
> |             | D2   | -0.0001 | **0.0000** | 0.0000  | 0.0003  | 0.0046  | 0.0001  | 0.0010  |
> |             | D3   | 0.0000  | **0.0000** | 0.0000  | 0       | 0.0002  | 0.0063  | 0.0063  |
> |             | D4   | 0.0000  | **0.0000** | 0.0000  | -0.0003 | 0.0005  | 0.0011  | 0.0021  |
> |             | D5   | 0.0000  | **0.0000** | 0.0000  | -0.0001 | 0.0006  | 0.0046  | 0.0033  |
>
> Robustness:
> | Method    | D1      | D2      | D3      | D4      | D5      |
> |:----------|---------|---------|---------|---------|---------|
> | GradCAM   | 0.9987  | 0.9949  | 0.9900  | 0.9908  | 0.9866  |
> | GOAT      | 0.9934  | 0.9927  | 0.8350  | 0.9852  | 0.9530  |
> | Ours      | 0.9774  | 0.9800  | 0.7820  | 0.9596  | 0.8576  |
> | IG        | 0.9627  | 0.9545  | 0.2728  | 0.9791  | 0.2493  |
> | GNN-LRP   | 0.6553  | 0.6588  | 0.1929  | 0.5810  | 0.1267  |
> | Random    | 0.5089  | 0.5087  | 0.5119  | 0.5229  | 0.5149  |
> | LIME      | 0.3879  | 0.3962  | 0       | 0.0270  | 0.0072  |
>
> More datasets:
>
> | ID | Dataset    | Graphs | Nodes   | Edges    | Features | Classes |
> |----|:-----------|--------|--------:|---------:|---------:|--------:|
> | D1 | Computers  | 1      | 13,752  | 491,722  | 767      | 10      |
> | D2 | Photo      | 1      | 7,650   | 238,162  | 745      | 8       |
> | D3 | CiteSeer   | 1      | 3,327   | 9,104    | 3,703    | 6       |
> | D4 | PubMed     | 1      | 19,717  | 88,648   | 500      | 3       |
> | D5 | Cora       | 1      | 2,708   | 10,556   | 1,433    | 7       |
>
> **W3**: We thank the reviewer for the feedback. We have tried our approach on more datasets (Computer, Photo, CiteSeer, PubMed and Cora), please check our response in the previous comment.

---

### Official Review · Reviewer_9dA9 · 2025-10-31

**Soundness:** 2
**Presentation:** 2
**Contribution:** 1
**Rating:** 2
**Confidence:** 4

**Summary:**

This paper proposes a decomposition framework for explaining GNN embeddings by attributing each embedding dimension to original node and edge features. The core idea is to treat GNN layers as linear operators (after fixing activation patterns) and propagate contribution matrices backward through the network in parallel with forward pass.

The paper addresses a relevant problem with a mathematically sound approach, but has significant weaknesses in evaluation methodology, technical novelty, and experimental validation. The lack of quantitative metrics and baseline comparisons is particularly concerning for a paper claiming to provide "trustworthy" explanations. Major revisions addressing these issues would be necessary before the paper meets standards for publication at a top-tier venue.

**Strengths:**

S1: The paper addresses a relevant gap in GNN explainability literature. When GNN embeddings are used as engineered features in downstream models, understanding what information these embeddings capture becomes critical

S2: The framework is applied to two different GNN architectures and two different tasks, showing that the core idea can be adapted to different settings.

S3: The method is computationally efficient. It reduces to simple matrix multiplications and does not require optimization, sampling, or iterative procedures.

**Weaknesses:**

W1: The paper's most critical limitation is absence of quantitative evaluation of explanation quality. All main results (Figures 1, 2) are purely qualitative visualizations without numerical metrics. There is no formal definition of what constitutes a "good" or "faithful" explanation, no metrics to measure correctness of attributions, and no way to objectively compare explanation quality across methods. The only quantitative experiment (Section 3.3) measures noise feature suppression, which is insufficient to validate the method comprehensively. For a paper claiming to provide "trustworthy" explanations, this is a significant gap.

W2: The paper does not compare against straightforward adaptations of gradient-based explanation methods (e.g., Integrated Gradients, GradCAM) for explaining embeddings. Computing $\frac{\partial h^{(L)}_v[d]}{\partial x_w[p]}$ for each embedding dimension $d$ with respect to input features would provide conceptually similar attributions. Without this comparison, it is unclear what advantages the proposed method offers beyond standard gradient-based approaches. The comparison in Section 3.3 uses methods designed for explaining predictions, not embeddings, which makes comparison unfair.

W3: The core technical contribution (Section 2.1) is essentially standard application of chain rule for backpropagating through linear transformations with ReLU activations. The propagation equation is straightforward linear algebra. The extensions to GCN and HinSAGE (Sections 2.2-2.3) are direct applications without significant innovation.

W4: 	Several highly relevant works on embedding interpretability are missing:
- Piaggesi et al. "Dine: Dimensional interpretability of node embeddings" (TKDE 2024)
- Shafi et al. "Generating human understandable explanations for node embeddings" (2024)
- Dalmia et al. "Towards interpretation of node embeddings" (WWW 2018)
- Piaggesi et al. "Disentangled and Self-Explainable Node Representation Learning" (TMLR 2025)
These works contain relevant metrics for quantitative evaluation that could strengthen this paper.

**Questions:**

See weaknesses.

---

> ### Author Response · Authors · 2025-11-21
>
> **W1**: We thank the reviewer for highlighting this critical point regarding quantitative evaluation. We have conducted additional experiments using widely adopted metrics for explanation faithfulness and robustness such as Fidelity +, Fidelity – and Robustness. We compared six other approaches on five datasets.
>
> | Metric       | Data | GOAT    | Ours    | GNN-LRP | IG      | GradCAM | LIME    | Random  |
> |:------------|------|---------|---------|---------|---------|---------|---------|---------|
> | Fidelity₊   | D1   | 0.0352  | **0.0353** | 0.0288  | 0.0164  | 0.0201  | 0.0070  | 0.0013  |
> |             | D2   | 0.0340  | **0.0342** | 0.0227  | 0.0164  | 0.0024  | 0.0114  | 0.0009  |
> |             | D3   | 0.1523  | **0.1514** | 0.1513  | 0.1239  | 0.1114  | 0.0281  | 0.0055  |
> |             | D4   | 0.0389  | **0.0384** | 0.0313  | 0.0196  | 0.0215  | 0.0010  | 0.0010  |
> |             | D5   | 0.1808  | **0.1789** | 0.1787  | 0.1229  | 0.1073  | 0.0267  | 0.0054  |
> | Fidelity₋   | D1   | -0.0001 | **0.0000** | 0.0000  | -0.0002 | 0.0002  | 0.0004  | 0.0013  |
> |             | D2   | -0.0001 | **0.0000** | 0.0000  | 0.0003  | 0.0046  | 0.0001  | 0.0010  |
> |             | D3   | 0.0000  | **0.0000** | 0.0000  | 0       | 0.0002  | 0.0063  | 0.0063  |
> |             | D4   | 0.0000  | **0.0000** | 0.0000  | -0.0003 | 0.0005  | 0.0011  | 0.0021  |
> |             | D5   | 0.0000  | **0.0000** | 0.0000  | -0.0001 | 0.0006  | 0.0046  | 0.0033  |
>
> Robustness:
> | Method    | D1      | D2      | D3      | D4      | D5      |
> |:----------|---------|---------|---------|---------|---------|
> | GradCAM   | 0.9987  | 0.9949  | 0.9900  | 0.9908  | 0.9866  |
> | GOAT      | 0.9934  | 0.9927  | 0.8350  | 0.9852  | 0.9530  |
> | Ours      | 0.9774  | 0.9800  | 0.7820  | 0.9596  | 0.8576  |
> | IG        | 0.9627  | 0.9545  | 0.2728  | 0.9791  | 0.2493  |
> | GNN-LRP   | 0.6553  | 0.6588  | 0.1929  | 0.5810  | 0.1267  |
> | Random    | 0.5089  | 0.5087  | 0.5119  | 0.5229  | 0.5149  |
> | LIME      | 0.3879  | 0.3962  | 0       | 0.0270  | 0.0072  |
>
> **W2**: We thank the reviewer for the helpful suggestion. We have added both Integrated Gradients and GradCAM in the quantitative experiments. Please see the results above in W1.
>
> **W3**: We thank the reviewer for this thoughtful observation. The underlying mathematics—primarily the application of the chain rule through linear transformations with piecewise-linear activations—is standard and widely used in backpropagation. The novelty of our work lies in the framework and its application to GNN interpretability, rather than in the derivation itself.
>
> **W4**: We thank the reviewer for pointing out these highly relevant works. We acknowledge that the works are important contributions to embedding interpretability and should have been included. We have added and discussed them in the revised manuscript.

---

> > ### Comment · Reviewer_9dA9 · 2025-11-24
> >
> > Thank you for the additional experiments. However, I have significant concerns that need clarification before I can assess whether weaknesses are adequately addressed.
> >
> > Your method's results are nearly identical to GOAt. This suggests both methods compute essentially the same quantities. Please clarify: Is your method an adaptation of GOAt applied to embeddings rather than predictions? What specific differences exist between your approach and GOAt for embedding explanation?
> >
> > The results table lacks critical details and interpretation. Please provide: (1) definitions of how Fidelity+/Fidelity-/Robustness were computed for embeddings specifically (standard definitions are for predictions), (2) how IG and GradCAM were adapted for embeddings, and (3) discussion of the results: why is Robustness consistently lower for your method than GOAt/IG? What explains the large metric variations across datasets? More importantly, are these three metrics sufficient for evaluating embedding explainability? They measure stability and faithfulness to model behavior, but do they capture whether explanations are actually interpretable or useful for understanding what information embeddings encode?

---

> > > ### Author Response · Authors · 2025-11-29
> > >
> > > We sincerely appreciate the reviewer's additional constructive and insightful comments.
> > >
> > > **Q1: Your method's results are nearly identical to GOAt. This suggests both methods compute essentially the same quantities. Please clarify: Is your method an adaptation of GOAt applied to embeddings rather than predictions? What specific differences exist between your approach and GOAt for embedding explanation?**
> > >
> > >
> > > No. Our method is an exact layer-wise decomposition of the 2-layer GCN mapping, which we can apply to either hidden embeddings or logits. When we apply it to the final logit layer, GOAt’s feature attributions are recovered as the special case obtained by summing our $ϕ_{v,u,f}^{decompose}$ over all source nodes $u$, where $ϕ_{v,u,f}^{decompose}$ is the contribution that the feature dimension f of node u makes node v. For the specific 2-layer GCN, this exact factorization happens to coincide with GOAT once you marginalize over source nodes, but our formulation is designed to generalize beyond this architecture (e.g., to our embedding modules with multiple streams).
> > > The main differences are: Firstly, our method returns a full tensor $ϕ_{v,u,f}^{decompose}$ (target node $v$, source node $u$, feature $f$). This lets us separate self contributions from n-hop contributions, and,  moreover, attribute the effects to specific neighbor types or streams (e.g., account vs merchant vs self nodes in our broader setting.) GOAt, however, directly computes $ϕ_{v,f}^{GOAt}$, already summed over all neighbors. It does not track which neighbor contributed what; the node-wise structure is lost. Second, our decomposition extends to multi-stream architectures (e.g., HinSAGE) where the node representation is formed by combining several modules (e.g., self features, merchant features, account history, etc.). Because we keep track of $ϕ_{v,u,f}^{decompose}$ at the module/input level, we can interpret contributions per stream and per hop data as a single monolithic feature vector; GOAt does not distinguish which stream or module produced a given dimension. Lastly, our method is derived as an exact algebraic factorization of the GCN forward pass into adjacency factors, ReLU masks, and weight matrices, with a strict conservation property by construction. GOAt is originally motivated as a Taylor-style attribution around the input used for prediction(fixed ReLU mask).
> > >
> > >
> > > **Q2: The results table lacks critical details and interpretation. Please provide: (1) definitions of how Fidelity+/Fidelity-/Robustness were computed for embeddings specifically (standard definitions are for predictions)**
> > >
> > > We appreciate the reviewer’s insightful comment. Although the metrics are traditionally defined for prediction-level explanations, our adaptation preserves their core meaning: mask the features an explanation marks as important and test whether the model’s predicted probability drops accordingly. By using the metrics, we measure whether the explanation selects important node features whose contributions propagate into the embedding and subsequently affect the prediction. We would like to provide more details about the definitions of the metrics:
> > >
> > > For a node $v$, let $p_v$ denote the softmax probability of its target class and let $s_v ∈ R^D$ be the explanation scores over the $D$ embedding dimensions. For a given $k$, we denote by $S_v^{top}(k)$ and $S_v^{bottom}(k)$ the indices of the $k$ largest and smallest entries of $s_v$ , respectively. Fidelity+ is defined as the average drop in target-class probability when we zero out the top-k embedding dimensions: $Fidelity^+(k) = E_v [p_v - p_v^{-S_v^{top}(k)}]$.Fidelity− is defined analogously by zeroing the bottom-k dimensions, which measures the drop when only the top-k dimensions are retained, and all others are set to zero: $Fidelity^-(k) = E_v [p_v - p_v^{-S_v^{bottom}(k)}]$. For Robustness, let $s_v^{clean}$ and $s_v^{noisy}$ be the embedding-level feature importance scores obtained on clean and noisy inputs, where noisy data were generated by adding normal noise to the original input, respectively. For a given $k$, we select the top-k dimensions $S_v^{top}(k)$ according to $s_v^{clean}$ and compute: $Robustness(k) = E_{v,d \\in S_v^{top}(k)}[1-(|s_{v,d}^{clean} - s_{v,d}^{noisy}|)/(|s_{v,d}^{clean}|+\epsilon)]$, which lies in [0,1] and quantifies the relative stability of the top-k embedding dimensions.

---

> > > ### Author Response · Authors · 2025-11-29
> > >
> > > **Q3: how IG and GradCAM were adapted for embeddings and discussion of the results: why is Robustness consistently lower for your method than GOAt/IG? What explains the large metric variations across datasets?**
> > >
> > > **IG**: We decompose the GNN into an encoder $f_{\\theta}$ and a classifier head $g_{\\theta}$. For a given node $v$, we first compute its embedding $h_v=f_{\\theta}(X)$ with $\\theta$ frozen. We then apply Integrated Gradients in embedding space: we choose a baseline embedding $h_v^{(0)}$ (we use the zero vector in our experiments) and integrate the gradient of the target logit with respect to the embedding along the straight-line path $h_v^{(\\alpha)}= h_v^{(0)}+\\alpha (h_v-h_v^{(0)}), \\alpha \\in [0,1]$, while keeping the encoder fixed. The resulting attribution for embedding dimension $d$ is $IG_{v,d}=(h_{v,d} - h_{v,d}^{(0)}) E_{\\alpha \\in [0,1]}[\\partial{g_{\\theta} (h_v^{(\\alpha)})}/\\partial{h_{v,d}^{(\\alpha)}}]$. **GradCAM**: We keep the encoder fixed, compute the node embeddings $h_v$, and then compute the gradient of the target logit with respect to each embedding coordinate: $g_{v,d} = \\partial{g_{\\theta} (h_v))}/\\partial{h_{v,d}}$. Following the GradCAM convention of using only positive influences, we define the importance of embedding dimension d for node $v$ as $CAM_{v,d}=max⁡(g_{v,d},0)$, yielding one attribution score per embedding dimension.
> > >
> > > Our decomposition method produces a much more fine-grained attribution map than GOAT or IG: instead of outputting a single feature score per node, it keeps per-neighbor, per-hop, per-feature contributions. This higher resolution makes the method more sensitive to small perturbations of the input. When we add noise to features, the exact message-passing paths and ReLU activation patterns change slightly, and because our method decomposes contributions down to each $\\phi_{v,u,f}$ these small changes produce larger relative differences in the final top-k scores. In contrast, GOAT and IG already collapse contributions (GOAT sums over neighbors; IG integrates over a smoothed path), making them inherently more stable. Thus, our method is more discriminative but also more sensitive, which leads to lower robustness values.
> > >
> > > In terms of the large metric variations, different datasets have different structural and feature properties, which affect explanation stability. For example, sparse, high-variance, low-homophily datasets (e.g., Cora) exhibit more unstable activation patterns under noise, resulting in low robustness for all methods. In contrast, dense continuous features and higher homophily in the Amazon datasets usually produce smoother gradients and more stable attributions, leading to relatively higher robustness values than Cora. These cross-dataset differences reflect intrinsic structural and statistical diversity rather than artifacts of the explanation methods

---

> > > ### Author Response · Authors · 2025-11-29
> > >
> > > **Q4: More importantly, are these three metrics sufficient for evaluating embedding explainability? They measure stability and faithfulness to model behavior, but do they capture whether explanations are actually interpretable or useful for understanding what information embeddings encode?**
> > >
> > > We agree that the selected metrics alone cannot fully characterize whether explanations are “interpretable” in a human sense. Our aim in this work is more modest and more mechanistic: we focus on how faithfully the explanations capture the model’s actual dependence on embedding dimensions, and how stable these explanations are under small perturbations. In this setting, the three metrics we use are natural and widely adopted quantitative proxies.
> > >
> > > For embeddings, Fidelity+ and keep-top answer exactly this question by measuring the effect of zeroing or retaining the top-k embedding coordinates on the target class probability. Fidelity− checks that dimensions marked as unimportant can be perturbed without significantly changing the prediction. Robustness then quantifies whether the ranking of embedding dimensions is stable under small input perturbations. Together, these metrics test whether our method provides faithful and stable attributions of embedding coordinates with respect to the trained model.
> > >
> > > We agree with the reviewer that these metrics do not directly assess whether the resulting explanations are semantically meaningful or useful to humans for understanding what information the embeddings encode. Our method is a local explanation method that decomposes, for each target node, how specific node features and neighbor contributions flow through the embedding to influence the prediction. This is evident in analyses such as our MovieLens experiment, where the explanation highlights how specific user-movie interactions contribute to embeddings. Therefore, addressing this question typically requires extra studies (e.g., predicting hand-crafted node attributes from the most important embedding coordinates). Designing such evaluations for graph embeddings is an interesting and non-trivial problem, and we view it as complementary to the present work. Due to scope constraints, we therefore focus here on model-centric faithfulness and stability and leave a systematic study of semantic interpretability of embedding explanations to future work.

---

### Official Review · Reviewer_KsJu · 2025-11-06

**Soundness:** 1
**Presentation:** 2
**Contribution:** 1
**Rating:** 0
**Confidence:** 4

**Summary:**

The authors present a framework aimed at explaining the embeddings of GNNs at the feature level, which is a clear task relevant for the community. The central premise is that existing GNN explainability methods focus on identifying important graph structures (nodes/edges) for a given prediction, but fail to decompose the learned embedding itself into contributions from the original input features. To address this, the paper proposes a method to "invert" GNN layers. The core idea is to treat a GNN layer as a linear operator once the outcomes of its non-linear activation functions are fixed for a specific input. By propagating "contribution matrices" backward through this linearized computational graph, the method claims to produce an exact decomposition of the final embedding vector into a sum of contributions from every input feature of every node in its computational subgraph. The approach is demonstrated on GCN and HinSAGE architectures using the Cora and MovieLens datasets.

**Strengths:**

The paper identifies an interesting and relevant problem within the GNN explainability literature. The distinction between explaining a final prediction and explaining the intermediate embedding representation is a valid one, and the goal of achieving feature-level attribution is desirable. The authors additionally frame the vast literature about explainability for GNN, comparing some popular methods with each other (Table 1), and showing potential gaps and future research directions.

**Weaknesses:**

Despite the interesting problem statement, the paper's methodology, claims, and experimental validation suffer from fundamental flaws, lack of clarity, and are of insufficient quality for publication at a top-tier conference as ICLR.
Here are some points that could be addressed by the authors to improve their work:

- The central and most critical weakness lies in the claim of providing an "exact decomposition" of the GNN's embedding. The method does not explain the true GNN model; instead it explains a linearized surrogate of the model that is valid only for a single input instance. The core of the method is to replace the non-linear activation function with a diagonal gating matrix $D$ (not formally defined). While the decomposition is "exact" for this new linear model, this is not the GNN that made the prediction. The non-linearity is a crucial component of the GNN's expressive power (or any Neural Network); hence, by neglecting it, the authors are analyzing a totally different function. This all raises severe questions about the method's faithfulness to the original model's learned representations and its overall justification.
Additionally, in Sec 4.1 the authors admit that this method achieves "exactness only for specific activation and normalization classes.", i.e., piece-wise linear activations, relegating the contribution of this paper to a narrow set of architectures. Therefore, the proposed scope of a general framework for feature-level explanation of GNN is rather a misleading characterization, as well as an overstatement.
- The experimental evaluation is limited to two specific datasets, Cora and Movielens. These are small and outdated benchmark datasets that do not test the efficacy of an explanation method in complex scenarios, or with larger graphs, or with higher-dimensional features and more subtle feature interactions. There are many other benchmarks, also for XAI, that should be considered for an impactful publication, which are employed in the related works cited by the authors too. Also, the authors compare their embedding explainer against methods designed to explain predictions (like GNNExplainer), which feels like a mismatched comparison.
- The evaluation is largely qualitative and anecdotal. The t-SNE plots (Fig 1) are visually appealing but notoriously unreliable for drawing firm conclusions (it does not really show "clear separation" across categories). The case studies on MovieLens (Figures 2, 6, 7, 8) present "just-so" stories about specific embeddings and features. While illustrative, they lack a rigorous and quantitative validation, and it is unclear if these cherry-picked examples are representative or meaningful (What about the other embeddings?).
-  The method outputs a contribution matrix $C$ where $C[p, k]$ represents the contribution of input feature $p$ to the $k$-th dimension of the final embedding (for some given node). It is entirely unclear how a user is meant to interpret this. What does it mean for the "scaled_age" feature to contribute -0.21 to the 25th embedding dimension for "animation" movies? The paper attempts to solve this by using PCA to create aggregate scores, but this introduces yet another layer of approximation and obfuscation, as the principal components themselves often lack clear semantic meaning. The final explanations are not inherently interpretable and require significant post-hoc processing that further distances them from the original model.
- The code is not provided, nor as supplementary material nor in an anonymized repo, hence it is impossible to reproduce the results and make this work less trustworthy.

Minor comments that did not affect the score:
- Although the part on the related work is clear and well-written, many arXiv references should be avoided as they are not peer-reviewed scientific contributions. Please refer to published versions when available, or avoid referencing arXiv unless strictly necessary.
- Sec. 2.1 on the definition of a GNN is not correct. GNNs entail a broad class of methods, and what you describe is not a "generic" GNN, but rather an instance of the message-passing relational graph convolutional operator (e.g., Schlichtkrull 2017). This should be properly defined.
- The quality of figures and their labels do not meet publication standards (e.g., Fig. 2 has fonts too small).
- Section 4.2 about Future works is inconclusive. What other advancements are possible beyond implementing this work in other convolutions?
- The statement "our proposed attribution-based decomposition not only provides interpretability for graph-based embeddings but also uncovers actionable insights into how heterogeneous relational signals shape predictive performance, thereby bridging the gap between representation learning and model explainability." is too exaggerated. You reported one simple example, prone to confirmation bias, that does not "bridge the gap between representation learning and model explainability", nor theoretically nor experimentally.

**Questions:**

- How can you claim the method is "exact" and faithful when it analyzes a linearized and input-dependent (PCA) surrogate model? The function being explained is fundamentally different from the GNN used for prediction. Could you please clarify this discrepancy?
- How is the " $D_v^{(l)}$ Diagonal mask from activation of node $v$ at layer $ℓ$" defined? It is a relevant component of your framework, but it is not obvious and never defined.
- The experimental validation relies heavily on qualitative case studies and a single synthetic downstream task. Why were established quantitative explainability metrics (e.g., Fidelity, Sparsity, Robustness) not used to provide a more rigorous comparison against baseline methods? What is the reason behind using only 140 for training in Cora? Did you cross-validate your results across other data splits and multiple runs?
- You are using XGBoost for downstream tasks on GNN embeddings instead of the original high-dimensional data embeddings. You retrieve the feature contributions via a PCA decomposition. Could you please try to apply PCA on Cora and then XGBoost without the GNN part? This ablation study is missing, but it would be essential to pinpoint the effects of the GNN (and its embeddings).
- How is it possible that your method is an order of magnitude faster than a random explainer that does not perform any computation at all?

---

> ### Author Response · Authors · 2025-11-21
>
> **W1**: Thank you for the constructive comments. Our method does not approximate the GNN with a surrogate model across the entire network. Instead, we observe that when the activation function is piecewise-linear (e.g., ReLU, LeakyReLU), the GNN’s predictions are locally linear, and thus we can provide an exact decomposition of the original message-passing GNN for the given input. The diagonal gating matrix D, which we have defined more formally in the revision ($D_v^{\ell}$ is a diagonal matrix that encodes the activation pattern for the given input for node v at layer $\ell$.), captures the activation state of each neuron for that input. This means the non-linear effect is preserved because the gating pattern is derived from the actual forward pass of the original GNN. The decomposition is not a simplification—it is a structural re-expression of the same computation graph under the activation pattern induced by the input.
>
> We agree that non-linearity is crucial for GNN expressiveness. Our approach does not discard it; rather, it leverages the fact that piecewise-linear activations (such as ReLU and LeakyReLU) partition the input space into regions where the GNN behaves as a linear operator. Within such a region, the decomposition is mathematically exact because the activation pattern is fixed for the given input instance. This property of ReLU-based networks—being piecewise linear—allows us to preserve the non-linear effect while enabling an exact attribution decomposition for that activation region.
>
> We acknowledge that while the method is exact for piecewise-linear activations and normalization schemes that preserve linearity within regions, this class includes a large number of widely used GNN architectures (GCN, GraphSAGE, GAT with ReLU), and thus still useful. We have revised the manuscript to clarify this scope and to avoid overstating the generality. Our contribution is to provide a principled decomposition for these architectures, which dominate practical applications.
>
> **W2**: Thank you for the insightful comments. We have addressed these concerns in the revised manuscript: We expanded dataset coverage, and included more datasets (e.g., Amazon - computer/photo), which have larger graphs and higher-dimensional node features.
>
> | ID | Dataset    | Graphs | Nodes   | Edges    | Features | Classes |
> |----|:-----------|--------|--------:|---------:|---------:|--------:|
> | D1 | Computers  | 1      | 13,752  | 491,722  | 767      | 10      |
> | D2 | Photo      | 1      | 7,650   | 238,162  | 745      | 8       |
> | D3 | CiteSeer   | 1      | 3,327   | 9,104    | 3,703    | 6       |
> | D4 | PubMed     | 1      | 19,717  | 88,648   | 500      | 3       |
> | D5 | Cora       | 1      | 2,708   | 10,556   | 1,433    | 7       |
>
> We acknowledge that comparing embedding-level explanations to prediction-level explainers (e.g., GNNExplainer) was not ideal. In the revised version, we remove GNNExplainer and instead include two methods that are more appropriate for embedding interpretation: GNN-LRP and GradCAM.

---

> ### Author Response · Authors · 2025-11-21
>
> **W3** : We thank the reviewer for the constructive and insightful feedback. We agree that t-SNE plots can be visually appealing but are not sufficient for drawing firm conclusions. Our intention was not to claim strong separation based solely on t-SNE but to provide an intuitive illustration of how decomposed embedding contribution values behave under our framework. We revised the text to clarify this point. In the revised manuscript, "Because this analysis is conducted at the feature-contribution level, it offers transparency into what each embedding learns individually by grouping features with similar contribution behaviors, helping interpret the role of neighborhood aggregation in shaping model predictions."
>
> The case studies on MovieLens were included to demonstrate how practitioners could use our method in real-world applications for feature-level interpretation. These examples were meant to be illustrative rather than exhaustive, and to provide some overall intuition about the overall patterns of the local explanations. To address the concern about representativeness, we have included other examples in the Appendix.
>
> We appreciate the suggestion for more rigorous evaluation. In response, we have conducted additional quantitative experiments using widely adopted metrics for explanation quality: Fidelity+, Fidelity-, and Robustness, comparing our method against existing approaches. These results are now in revised manuscript.
>
> Fidelity+ and Fidelity-:
> | Metric       | Data | GOAT    | Ours    | GNN-LRP | IG      | GradCAM | LIME    | Random  |
> |:------------|------|---------|---------|---------|---------|---------|---------|---------|
> | Fidelity₊   | D1   | 0.0352  | **0.0353** | 0.0288  | 0.0164  | 0.0201  | 0.0070  | 0.0013  |
> |             | D2   | 0.0340  | **0.0342** | 0.0227  | 0.0164  | 0.0024  | 0.0114  | 0.0009  |
> |             | D3   | 0.1523  | **0.1514** | 0.1513  | 0.1239  | 0.1114  | 0.0281  | 0.0055  |
> |             | D4   | 0.0389  | **0.0384** | 0.0313  | 0.0196  | 0.0215  | 0.0010  | 0.0010  |
> |             | D5   | 0.1808  | **0.1789** | 0.1787  | 0.1229  | 0.1073  | 0.0267  | 0.0054  |
> | Fidelity₋   | D1   | -0.0001 | **0.0000** | 0.0000  | -0.0002 | 0.0002  | 0.0004  | 0.0013  |
> |             | D2   | -0.0001 | **0.0000** | 0.0000  | 0.0003  | 0.0046  | 0.0001  | 0.0010  |
> |             | D3   | 0.0000  | **0.0000** | 0.0000  | 0       | 0.0002  | 0.0063  | 0.0063  |
> |             | D4   | 0.0000  | **0.0000** | 0.0000  | -0.0003 | 0.0005  | 0.0011  | 0.0021  |
> |             | D5   | 0.0000  | **0.0000** | 0.0000  | -0.0001 | 0.0006  | 0.0046  | 0.0033  |
>
> Robustness
> | Method    | D1      | D2      | D3      | D4      | D5      |
> |:----------|---------|---------|---------|---------|---------|
> | GradCAM   | 0.9987  | 0.9949  | 0.9900  | 0.9908  | 0.9866  |
> | GOAT      | 0.9934  | 0.9927  | 0.8350  | 0.9852  | 0.9530  |
> | Ours      | 0.9774  | 0.9800  | 0.7820  | 0.9596  | 0.8576  |
> | IG        | 0.9627  | 0.9545  | 0.2728  | 0.9791  | 0.2493  |
> | GNN-LRP   | 0.6553  | 0.6588  | 0.1929  | 0.5810  | 0.1267  |
> | Random    | 0.5089  | 0.5087  | 0.5119  | 0.5229  | 0.5149  |
> | LIME      | 0.3879  | 0.3962  | 0       | 0.0270  | 0.0072  |
>
> **W4**: We sincerely thank the reviewer for the insightful and constructive feedback on interpretability. The contribution matrix C is designed to provide fine-grained attribution at the feature level for each embedding dimension. While a single entry such as "scaled_age" seem abstract in isolation, the intent is to allow users to trace how input features influence specific latent dimensions. We agree that PCA introduces an additional layer of abstraction and that principal components may lack clear semantics. Our use of PCA was motivated by the need to aggregate high-dimensional contributions into a smaller set of interpretable axes for visualization purposes. Importantly, PCA is applied only as a post-hoc tool for visualization, not as part of the explanation mechanism itself. The raw contribution matrix remains available for users who prefer direct attribution analysis. We have added a discussion on this point and mentioned some alternative aggregation strategies.
>
> **W5**: We thank the reviewer for bringing this up. We have attached the code.

---

> ### Author Response · Authors · 2025-11-21
>
> **Minor1**: We greatly appreciate the reviewer’s attention to detail and valuable suggestions regarding citation practices. We have replaced arXiv citations with their corresponding peer-reviewed versions where available (e.g., conference or journal publications).
>
> **Minor 2**:
> We thank the reviewer for the careful observation and helpful suggestion regarding the definition of GNNs. We acknowledge that our initial description corresponds to the message-passing formulation, specifically the relational graph convolutional operator introduced by Schlichtkrull et al. (2017), rather than the entire class of GNN architectures. We have revised Sec. 2.1 to explicitly state that the definition refers to the message-passing paradigm, not a generic GNN.
>
> **Minor 3**: Thank you for pointing out this. We have improved the quality of some of the figures.
>
> **Minor 4**: We appreciate the reviewer’s very constructive suggestion regarding the future work section. In the revised manuscript, we expanded Section 4.2 (now 5.2) to include some potential advancements: Extending the framework to provide confidence measures for explanations under distribution shifts or adversarial perturbations.
>
> **Minor 5**: We thank the reviewer for highlighting this important point about the strength of our claims. We rephrased the statement to more accurately reflect the contribution, for example: “Our proposed attribution-based decomposition provides a step toward improving interpretability of graph-based embeddings and offers insights into how heterogeneous relational signals influence predictive performance.”  We also clarified that the example provided is illustrative.
>
> **Question 1**:  We thank the reviewer for raising this important point. Our method does not explain a PCA-based surrogate model nor a fundamentally different function from the original GNN. The decomposition is performed on the actual message-passing GNN under the activation pattern induced by the input. For piecewise-linear activations (e.g., ReLU), the GNN behaves as a linear operator within the corresponding activation region. The diagonal gating matrix D encodes this activation state, ensuring that non-linearity is preserved for the given input. Thus, the decomposition is mathematically exact for the original GNN computation in that region.
>
> The PCA is not part of the explanation mechanism—it is used only as an optional post-hoc visualization tool to aggregate high-dimensional contributions for interpretability. The raw contribution matrix remains available for direct analysis without PCA. We have clarified this distinction in the revision to avoid confusion.
>
> **Question 2**: Thank you for raising this important point. We have defined D explicitly in the revised manuscript. $D_v^{\ell}$ is a diagonal matrix that encodes the activation pattern for the given input for node v at layer $\ell$.
>
> **Question 3**: Thank you so much for the valuable feedback. We have conducted additional quantitative experiments using widely adopted metrics for explanation quality: Fidelity+, Fidelity-, and Robustness, comparing our method against existing approaches. (See results above). In terms of the training size in Cora, we adapted the Planetoid split used in Yang et al. (2016) and Shchur et al. (2018), who used 20 samples per class (thus 140 in total for 7 classes) for training. We also evaluated the GNN performance using alternative data splits, and observed only minor differences in the results.
>
> Reference:
>
> Z. Yang, W. W. Cohen, and R. Salakhutdinov. Revisiting semi-supervised learning with graph embeddings. ICML, 2016
> Oleksandr Shchur, Maximilian Mumme, Aleksandar Bojchevski, and Stephan Günnemann. Pitfalls of Graph Neural Network Evaluation. In NIPS workshop, 2018

---

> ### Author Response · Authors · 2025-11-21
>
> **Question4**: Thank you for the thoughtful suggestion. We conducted the proposed experiment by applying PCA directly to the original feature space (Cora dataset) and then training XGBoost without using GNN embeddings. The XGB + PCA features don’t perform as well as XGB with graph embeddings. Please check the details below:
>
> XGB + PCA Features:
> | Class                     | Precision | Recall | F1-Score | Support |
> |---------------------------|-----------|--------|----------|---------|
> | Case_Based               | 0.59      | 0.62   | 0.60     | 227     |
> | Genetic_Algorithms       | 0.58      | 0.69   | 0.63     | 319     |
> | Neural_Networks          | 0.65      | 0.77   | 0.70     | 625     |
> | Probabilistic_Methods    | 0.72      | 0.42   | 0.53     | 325     |
> | Reinforcement_Learning   | 0.66      | 0.40   | 0.50     | 166     |
> | Rule_Learning            | 0.53      | 0.46   | 0.49     | 138     |
> | Theory                   | 0.44      | 0.50   | 0.47     | 268     |
> | **Accuracy**             |           |        | 0.60     | 2068    |
> | **Macro Avg**            | 0.60      | 0.55   | 0.56     | 2068    |
> | **Weighted Avg**         | 0.61      | 0.60   | 0.59     | 2068    |
>
> XGB + GNN Embeddings:
> | Class                     | Precision | Recall | F1-Score | Support |
> |---------------------------|-----------|--------|----------|---------|
> | Case_Based               | 0.71      | 0.65   | 0.68     | 227     |
> | Genetic_Algorithms       | 0.88      | 0.83   | 0.86     | 319     |
> | Neural_Networks          | 0.73      | 0.91   | 0.81     | 625     |
> | Probabilistic_Methods    | 0.89      | 0.65   | 0.75     | 325     |
> | Reinforcement_Learning   | 0.63      | 0.72   | 0.67     | 166     |
> | Rule_Learning            | 0.78      | 0.56   | 0.65     | 138     |
> | Theory                   | 0.67      | 0.64   | 0.65     | 268     |
> | **Accuracy**             |           |        | 0.76     | 2068    |
> | **Macro Avg**            | 0.76      | 0.71   | 0.73     | 2068    |
>
> **Question 5**: We thank the reviewer for pointing out this. The original timing reported in the paper only measured the matrix multiplication step of our decomposition method (and was divided by the number of observations), not the entire pipeline. In contrast, the random explainer was invoked through the PyTorch Explainer API, which introduces additional overhead (e.g., graph sampling, wrapper calls) even though the attribution itself is random. This led to an unfair comparison. We have reimplemented the random explainer to remove unnecessary latency and ensure a fair comparison. We now report the full pipeline time for both methods, including preprocessing and attribution steps. (Results (in second) were based on 100 nodes across different datasets).
>
> | Method   | D1       | D2       | D3       | D4       | D5       |
> |----------|---------:|--------:|---------:|---------:|---------:|
> | Random   | 0.040    | 0.023    | 0.047    | 0.038    | 0.017    |
> | GradCAM  | 0.132    | 0.071    | 0.058    | 0.065    | 0.021    |
> | GNN-LRP  | 0.327    | 0.535    | 0.308    | 0.535    | 0.269    |
> | Ours     | 45.782   | 13.857   | 0.193    | 0.749    | 0.486    |
> | GOAT     | 220.752  | 74.038   | 27.527   | 437.584  | 14.792   |
> | LIME     | 495.143  | 161.958  | 92.927   | 118.167  | 43.122   |
> | IG       | 611.970  | 252.356  | 194.829  | 224.865  | 56.606   |

---

### Author Response · Authors · 2025-12-03
**Rebuttal Summary to AC**

Dear AC,

Thank you for your time managing the review process and reviewing our submission.  We sincerely thank all reviewers for their time, effort, and constructive feedback, which has significantly helped improve the clarity and quality of our work. In response, we have revised the manuscript to broaden experimental evaluation, clarify the methodology, and strengthen interpretability and reproducibility, and refined the description of future work. Below is a summary of the main revisions:

**Expanded Experimental Evaluation (Reviewers  KsJu, 9dA9, mPHL, and 7hVM):** First, we expanded the experimental evaluation of explanation quality. Section 4 now includes quantitative evaluation results on larger and more diverse datasets such as Amazon-Computer, Amazon-Photo, PubMed, CiteSeer, and Cora. We evaluated explanation quality using widely adopted metrics—Fidelity+, Fidelity-, and Robustness—and compared our approach against six baselines, including two newly added methods, GNN-LRP and GradCAM, which were suggested by the reviewers and relevant for embedding-level interpretation. The results show that our method achieves state-of-the-art performance on Fidelity+ and Fidelity-, while maintaining competitive robustness. To further validate the design choices, we conducted an ablation study contrasting PCA + XGBoost with GNN embeddings + XGBoost, highlighting how graph embeddings improve downstream pattern learning.

**Clarification of Method and Scope (Reviewers KsJu  and 7hVM):** Second, we clarified the theoretical foundation and scope of our method. Our approach does not approximate the GNN with a surrogate model but instead provides an exact decomposition of the original message-passing GNN for the given input which is locally linear for certain activation functions. We also formally defined the diagonal gating matrix D and explained how it preserves non-linearity by encoding activation patterns from the original forward pass. To avoid overstating generality, we revised claims and rephrased some of the broader statements, ensuring that the scope is accurately described.

**Interpretability Enhancements (Reviewers KsJu and 7hVM):** We also improved interpretability and presentation. For qualitative experiments, we clarified that the t-SNE visualization in the Cora example is intended as an intuitive illustration rather than evidence of strong separation. For the MovieLens example, we explained how contribution values allow tracing input node features to latent embedding dimensions, even if individual contributions appear abstract in isolation. Regarding PCA, we clarified that its use was motivated by the need to aggregate high-dimensional contributions into interpretable axes for visualization and provided explicit formulas for PCA-based scoring methods ($s_{pca}$  and $s_{pca-var}$) to enhance transparency.

**Future Work and Additional Technical Clarifications (Reviewers KsJu, 9dA9 and  7hVM):** In addition, we expanded the future work section to include meaningful directions such as adapting our approach to graph transformers and incorporating confidence measures into explanations. We also provided additional technical clarifications and ensured reproducibility. We explained the relationship between our method and GOAt, emphasizing that our approach is an exact layer-wise decomposition generalizable beyond GCN architectures. We detailed how Fidelity+/Fidelity-/Robustness were computed for embeddings and how IG and GradCAM were adapted. We clarified that our goal is to interpret learned representations rather than training sample characteristics, which reflects an inherent property of inductive sampling-based GNNs. To ensure transparency, we provided train/test split protocols, dataset statistics (Appendix A.5, Table 4), and released the code. Image-based tables were replaced with proper LaTeX tables, and all experiments are fully reproducible.

---

### Meta-Review · Area_Chair_yupk · 2025-12-23

**Summary:**

The rebuttal of the authors has addressed a few concerns such as insufficient datasets, baselines, and quantitative evaluation. However, the following concerns haven't been well addressed: 1) overstatement of "exact reconstruction"; 2) the superiority of the proposed method over GOAT and GNN-LRP is not significant; 3) the problem formulation and technical contribution haven't sufficiently improved by the rebuttal and revision. Given these concerns, I have to recommend a rejection. I suggest that the authors make the following modification in the next version of the paper:
1. Improve the presentation
2. Use a chart to show the main idea of the proposed method
3. Reconsider the claims related to exact reconstruction
4. Provide some theoretical guarantees or justification, if possible
5. Provide more explanations for the mathematical formulas such as (2), (3), and those in line 245
6. Show the standard deviations in Table 2 and conduct significance tests

**Reviewer Concerns:**

**Reviewer KsJu**'s major concerns are:
1. The claim of providing an "exact decomposition" of the GNN's embedding is not supported by the proposed methods. Therefore, the proposed scope of a general framework for feature-level explanation of GNN is rather a misleading characterization, as well as an overstatement.
2. The experimental evaluation is limited to two specific datasets, Cora and Movielens.
3. The evaluation is largely qualitative and anecdotal.
4. The motivation of the contribution matrix hasn't been clearly explained.
5. The code is not provided, nor as supplementary material nor in an anonymized repo, hence it is impossible to reproduce the results and make this work less trustworthy.

I think that Concerns 2, 3, and 4 have been addressed by the rebuttal, Concern 5 is just a minor issue, and Concern 1 persists.

**Reviewer 9dA9**'s major concerns are:
1. The paper's most critical limitation is the absence of a quantitative evaluation of explanation quality.
2. The paper does not compare against straightforward adaptations of gradient-based explanation methods (e.g., Integrated Gradients, GradCAM) for explaining embeddings. It is unfair to compare methods designed for explaining predictions, not embeddings.
3. The core technical contribution is not significant.
4. Several highly relevant works on embedding interpretability are missing.

I think that Concerns 1, 2, and 4 have been addressed.

**Reviewer mPHL**'s major concerns are:
1. The difference and connection between the proposed method and GNN-LRP haven't been discussed.
2. More path-based explanation methods for GNNs, such as GNN-LRP, FlowX, and AxiomPath-Convex, should be compared in the experiments.
3. The experiments cover too few datasets.

Concerns 1 and 3 have been addressed by the rebuttal. However, the additional numerical results indicate that the improvement of the proposed method over the strongest competitors is not significant.

**Reviewer 7hVM**'s major concerns are:
1. Insufficient quantitative evidence and unclear evaluation protocol (e.g., missing details of data splitting, over-claiming "exact")
2. Over-claiming determinism and inconsistency
3. Reproducibility and reporting gaps
4. Writing and presentation issues

I think that Concerns 1 and 4 have been addressed by the rebuttal and Concerns 2 and 3 remain to some extent.

**Reviewer Scores:**

The initial ratings given by the four reviewers are either 0 or 2. If they had been able to participate fully in the discussion, they might have raised the ratings to 4, but not to 6, since a few concerns remain, and the additional numerical results showed that the improvement of the proposed method is not significant.

---

### Decision · Program_Chairs · 2026-01-26

Reject